# Consortium for the Study of Pregnancy Treatments (Co-OPT): An international birth cohort to study the effects of antenatal corticosteroids

Emily M. Frier[1]*, Chun Lin[2], Rebecca M. Reynolds[1,3], Karel Allegaert[4,5], Jasper V. Been[6], Abigail Fraser[7], Mika Gissler[8,9,10], Kristjana Einarsdóttir[11], Lani Florian[12], Bo Jacobsson[13,14,15], Joshua P. Vogel[16], Helga Zoega[11,17], Sohinee Bhattacharya[18], Eyal Krispin[19], Lars Henning Pedersen[20,21], Devender Roberts[22], Stefan Kuhle[23], John Fahey[24], Ben W. Mol[18,25], David Burgner[26,27], Ewoud Schuit[28], Aziz Sheikh[2], Rachael Wood[2,29], Cynthia Gyamfi-Bannerman[30], Jessica E. Miller[26,27], Kate Duhig[31], Marius Lahti-Pulkkinen[1,8,32], Eran Hadar[33,34], John Wright[35], Sarah R. Murray[1], Sarah J. Stock[2]

1 MRC Centre for Reproductive Health, The Queen's Medical Research Institute, The University of Edinburgh, Edinburgh, United Kingdom, 2 Usher Institute, The University of Edinburgh, Edinburgh, United Kingdom, 3 Centre for Cardiovascular Science, The University of Edinburgh, Edinburgh, United Kingdom, 4 Department of Development and Regeneration & Department of Pharmaceutical and Pharmacological Sciences, KU Leuven, Leuven, Belgium, 5 Department of Hospital Pharmacy, Erasmus University Medical Center Rotterdam, Rotterdam, The Netherlands, 6 Division of Neonatology, Department of Paediatrics / Department of Obstetrics and Gynaecology / Department of Public Health, Erasmus MC Sophia Children's Hospital, University Medical Centre Rotterdam, Rotterdam, The Netherlands, 7 Population Health Sciences, Bristol Medical School and MRC Integrative Epidemiology Unit at the University of Bristol, Bristol, United Kingdom, 8 THL Finnish Institute for Health and Welfare, Department of Knowledge Brokers, Helsinki, Finland, 9 Region Stockholm, Academic Primary Health Care Centre, Stockholm, Sweden, 10 Karolinska Institute, Department of Molecular Medicine and Surgery, Stockholm, Sweden, 11 Centre of Public Health Sciences, Faculty of Medicine, University of Iceland, Reykjavík, Iceland, 12 Moray House School of Education and Sport, University of Edinburgh, Edinburgh, United Kingdom, 13 Department of Obstetrics and Gynecology, Institute of Clinical Science, Sahlgrenska Academy, Gothenburg, Sweden, 14 Department of Obstetrics and Gynecology, Region Västra Götaland, Sahlgrenska University Hospital, Gothenburg, Sweden, 15 Department of Genetics and Bioinformatics, Domain of Health Data and Digitalization, Institute of Public Health, Oslo, Norway, 16 Maternal, Child and Adolescent Health Program, Burnet Institute, Melbourne, Australia, 17 School of Population Health, Faculty of Medicine & Health, University of New South Wales, Sydney, Australia, 18 Aberdeen Centre for Women's Health Research, Institute of Applied Health Sciences, School of Medicine, Medical Sciences and Nutrition, University of Aberdeen, Aberdeen, United Kingdom, 19 Boston Children's Hospital, Harvard University, Boston, Massachusetts, United States of America, 20 Department of Clinical Medicine, Aarhus University, Aarhus, Denmark, 21 Department of Obstetrics and Gynaecology, Aarhus University Hospital, Aarhus, Denmark, 22 Family Health Division, Liverpool Women's Hospital, Liverpool, United Kingdom, 23 Departments of Pediatrics and Obstetrics & Gynaecology, Dalhousie University, Halifax, Nova Scotia, Canada, 24 Reproductive Care Program of Nova Scotia, IWK Health, Halifax, Nova Scotia, Canada, 25 Ritchie Centre, Monash University, Clayton, Australia, 26 Murdoch Children's Research Institute, Royal Children's Hospital, Parkville, Victoria, Australia, 27 Department of Paediatrics, Melbourne University, Parkville, Victoria, Australia, 28 Julius Center for Health Sciences and Primary Care, University Medical Center Utrecht, Utrecht University, Utrecht, The Netherlands, 29 Public Health Scotland, Edinburgh, United Kingdom, 30 Department of Obstetrics, Gynecology and Reproductive Sciences, UC San Diego Health Sciences, La Jolla, California, United States of America, 31 Maternal and Fetal Health Research Centre, University of Manchester, Manchester, United Kingdom, 32 Department of Psychology and Logopedics, Faculty of Medicine, University of Helsinki, Helsinki, Finland, 33 Helen Schneider Hospital for Women, Rabin Medical Center, Petach-Tikva, Israel, 34 Sackler Faculty of Medicine, Tel-Aviv University, Tel-Aviv, Israel, 35 Bradford Institute for Health Research, Bradford Teaching Hospitals NHS Foundation Trust, Bradford, United Kingdom

* Emily.Frier@ed.ac.uk



**Data Availability Statement:** The Co-OPT ACS cohort is stored in the National Health Service

(NHS) Scotland National Safe Haven, provided by Public Health Scotland (PHS) electronic Data Research and Innovation Service (eDRIS). Patient-level data underlying this article cannot be shared publicly due to data protection and confidentiality requirements, as established by the individual data holders for data provided by each country, because the dataset contains sensitive and potentially identifying data. The Finnish National Institute for Health and Welfare, The Icelandic Directorate of Health, the Rabin Medical Center (Israel), The Reproductive Care Program of Nova Scotia, Public Health Scotland and National Records of Scotland are the data holders for the data used in this study. Data may be made available to approved researchers for analysis after securing relevant permissions from the data holders. Enquiries regarding data availability should be directed to Professor Mika Gissler, Finland (mika.gissler@thl. fi); Professor Kristjana Einarsdottir, Iceland (ke@hi. is); Dr Eyal Krispin, Israel (Eyalkrispin@gmail. com); Dr Stefan Kuhle, Canada (stefan.kuhle@dal. ca) and the eDRIS team at Public Health Scotland (phs.edris@phs.scot).

**Funding:** The Co-OPT ACS study is funded through a Wellcome Trust Clinical Career Development Fellowship grant (Funding Reference number 209560/Z/17) awarded to Sarah J Stock. The funders had no role in study design, data collection, data analysis, decision to publish, or preparation of the manuscript. The Sponsor of the study is the University of Edinburgh (www.ed.ac. uk), Sponsor reference AC19119. For the purpose of open access, the author has applied a CC BY public copyright licence to any Author Accepted Manuscript version arising from this submission.

**Competing interests:** The authors have declared that no competing interests exist.

# Abstract

## Background

Antenatal corticosteroids (ACS) are widely prescribed to improve outcomes following preterm birth. Significant knowledge gaps surround their safety, long-term effects, optimal timing and dosage. Almost half of women given ACS give birth outside the "therapeutic window" and have not delivered over 7 days later. Overtreatment with ACS is a concern, as evidence accumulates of risks of unnecessary ACS exposure.

## Methods

The Consortium for the Study of Pregnancy Treatments (Co-OPT) was established to address research questions surrounding safety of medications in pregnancy. We created an international birth cohort containing information on ACS exposure and pregnancy and neonatal outcomes by combining data from four national/provincial birth registers and one hospital database, and follow-up through linked population-level data from death registers and electronic health records.

## Results and discussion

The Co-OPT ACS cohort contains 2.28 million pregnancies and babies, born in Finland, Iceland, Israel, Canada and Scotland, between 1990 and 2019. Births from 22 to 45 weeks' gestation were included; 92.9% were at term ($\geq$ 37 completed weeks). 3.6% of babies were exposed to ACS (67.0% and 77.9% of singleton and multiple births before 34 weeks, respectively). Rates of ACS exposure increased across the study period. Of all ACS-exposed babies, 26.8% were born at term. Longitudinal childhood data were available for 1.64 million live births. Follow-up includes diagnoses of a range of physical and mental disorders from the Finnish Hospital Register, diagnoses of mental, behavioural, and neurodevelopmental disorders from the Icelandic Patient Registers, and preschool reviews from the Scottish Child Health Surveillance Programme. The Co-OPT ACS cohort is the largest international birth cohort to date with data on ACS exposure and maternal, perinatal and childhood outcomes. Its large scale will enable assessment of important rare outcomes such as perinatal mortality, and comprehensive evaluation of the short- and long-term safety and efficacy of ACS.

## Introduction

The Consortium for the Study of Pregnancy Treatments (Co-OPT) collaboration was created in 2018 to address the major gaps in knowledge surrounding the safety of medications in pregnancy, with the first Co-OPT study focusing on antenatal corticosteroids (ACS). The Consortium itself includes collaborators contributing data to the cohort, experts from complementary disciplines with methodological skills, and public and stakeholder representation.

Over the last 30 years, ACS have become an essential part of care for women considered to be at risk of imminent preterm birth (PTB) [1]. Liggins' and Howie's landmark randomised controlled trial (RCT) in 1972 demonstrated that ACS improved neonatal mortality

and respiratory distress syndrome (RDS) from PTB [2]. The 2020 Cochrane systematic review on ACS presented robust evidence that timely administration of ACS to women prior to early PTB reduces the most significant complications of PTB (neonatal mortality, RDS, and intraventricular haemorrhage), regardless of resource setting [1]. However, the majority of RCTs in this review represent only a subset of women who are offered ACS in clinical practice [1]. The population of women given ACS has expanded over time and ranges from women at 22 to 38 weeks' gestation [3, 4], with singleton or multiple pregnancies, even when the absolute risk of impending delivery is relatively low [5, 6]. Additionally, 15 trials in this Cochrane review were undertaken over 25 years ago [1], during a different era of neonatal care. This limits extrapolation of results to the current context of modern neonatal respiratory care and the associated improved outcomes for preterm infants in high-income countries [7]. Further research is required to define the risks and benefits of ACS in different obstetric populations.

Exposure to ACS is not without risk. Animal studies have shown a dose-dependent association of corticosteroid administration in late pregnancy with reduced fetal brain growth and delayed myelination [8]. Cohort studies in Finland and Canada demonstrated significant associations between *in utero* corticosteroid exposure with increased risks of mental and behavioural disorders [9], and increased healthcare utilisation for suspected neurodevelopmental disorders in childhood for term-born infants [10]. Animal models and cohort studies have also raised concerns regarding potential association of ACS exposure and adverse effects on cardiovascular, metabolic, renal and immune systems [4, 11, 12], and, particularly in term-born babies, reduced size at birth [13, 14]. As potential risks associated with ACS exposure emerge, further research is needed to improve understanding of the safety and benefits of ACS.

Furthermore, the impact of the treatment-to-delivery interval on the effects of ACS remains uncertain, and studies have reported mixed findings [15]. Although the therapeutic benefit of ACS is generally considered to be maximal in babies born within 24–48 hours after its administration [16], and likely dissipates after seven days [16–18], approximately 40–80% of women given ACS remain undelivered after this putative therapeutic window [19–21]. Given that the timing of ACS administration remains suboptimal, "overtreatment" with ACS is a concern.

The Co-OPT ACS cohort is a robust, international database of pregnant women and babies, designed to address research questions surrounding the benefits and safety of ACS. Through record linkage, Co-OPT amalgamates a variety of source data from birth and death registers, prescription databases, and electronic health records, which will be used to evaluate the impact of ACS on perinatal mortality and morbidity, maternal mortality and morbidity, and child health outcomes, including markers of neurodevelopment. In addition, Co-OPT aims to identify clinical features that influence maternal and infant outcomes following ACS exposure, and ultimately, to develop predictive models to optimise the prescription of ACS.

In this paper, we describe how we have linked population-based data sources from several international regions to create the Co-OPT ACS cohort, which will enable us to examine the effect of ACS exposure on maternal, perinatal and childhood outcomes. We also present the characteristics of the cohort and we consider its strengths and limitations.

## Materials and methods

### Protocol and registration

The Co-OPT ACS study protocol was registered in advance with PROSPERO (CRD42019137260) on 8th August 2019 [22].

## Overview of the Co-OPT ACS cohort

The Co-OPT ACS cohort is an international cohort of births from four prospectively collected national/provincial birth registers (Finland, Iceland, Nova Scotia (Canada) and Scotland) and one retrospectively collected hospital database (Rabin Medical Center, Israel). Births from 1990 onwards are included, as global uptake of ACS use increased significantly during the 1990s [23]. Live births from 22 to 45 weeks of gestation are included in the cohort. Inclusion of births from 22 completed weeks reflects recommendations from the World Health Organization to include all fetuses and infants which reach a gestational age of 22 weeks in national statistics [24]. Global definitions for stillbirths (death of baby before or during labour) vary; stillbirths are included in the cohort based on criteria used in the country of birth: the gestational threshold to record a stillbirth is 20 weeks in Israel and Nova Scotia, 22 weeks in Finland and Iceland, and 24 weeks in Scotland.

All countries and regions included in the cohort are classified as having high-income economies by the World Bank [25], and all regions provide public-funded maternity care; in Israel, approximately 20% of women opt for private antenatal care, although all births take place in public hospitals, such as the Rabin Medical Center.

The clinical guidelines on ACS which were in use in each region during the years when births in the cohort occurred are summarised in S2 Table. These were similar across countries within the Co-OPT ACS cohort and were consistent with recommendations from the World Health Organization [16].

## Data sources

Table 1 provides detailed information on all data sources contributing to the Co-OPT ACS cohort and includes information on annual numbers of live births and levels of neonatal care provided within each region. Variables available in the Co-OPT ACS cohort, along with their availability in the contributing datasets, are shown in Table 2. Further information on variables available from all regions after harmonisation of datasets is available in S1 Table. Detailed descriptions and definitions of all variables are provided in the Co-OPT ACS data dictionary, which is located at https://datashare.ed.ac.uk/handle/10283/4768.

**Maternity data.** Finnish and Icelandic Medical Birth Registers include data on all hospital births, planned home births (<0.2% of births in Finland [26] and 2% of births in Iceland [27]), planned births at midwives' clinics in Iceland (2% of Icelandic births), and unplanned births outside of the hospital. The high quality of the registers and their virtually complete coverage of all births has been described previously [28, 29]. The data in these registers are typically provided by midwives or physicians attending out-of-hospital births, or by the hospitals where women have given birth or where women been treated after the birth. In Finland, neonatal care for babies born before 32 weeks is centralised to five university hospitals. In Iceland, a neonatal intensive care unit at the university hospital in Reykjavík provides care for babies born after 22 weeks, and a special care unit in Akureyri also provides care for babies born after 34 weeks.

The Nova Scotia Atlee Perinatal Database contains data from all births in Nova Scotia and includes out-of-hospital births (1% of births in Nova Scotia), both planned and unplanned. Validation studies have demonstrated its high data quality [30]. The type of neonatal care provided for babies in Nova Scotia is region-dependent.

In Scotland, the Maternity Inpatient and Day Case Scottish Morbidity Records (SMR02) include data from inpatient and outpatient hospital-based "episodes" (any interaction with the health care system, irrespective of duration), which are submitted to Public Health Scotland (PHS) when a woman is discharged from that episode of care. Delivery data accounts

Table 1. Data sources, numbers of births and level of neonatal care provided in each region in the Co-OPT ACS cohort.

| Region | Category | Data sources | Years available | Description of data |
|---|---|---|---|---|
| FINLAND | Pregnancy & birth | Finnish Medical Birth Register | 2006–2018 | Population birth register of all births in Finland. **Births included:** live births and stillbirths $\geq$ 22 weeks, or birthweight $\geq$ 500 grams. **Number of live births per year:** 46,500 [29] **Neonatal care:** <br>• Babies < 32 weeks: care centralised to five university hospitals <br>• Babies $\geq$ 32 weeks: care provided in all birth hospitals |
| | Secondary care | Finnish Hospital Register (Care Register for Health Care) | 2006–2018 | Episode-level data on all hospital inpatient care (public and private hospitals) and outpatient visits (public hospitals). Includes primary diagnoses (and secondary diagnoses if recorded). |
| ICELAND | Pregnancy & birth | Icelandic Medical Birth Register | 2003–2017 | Population birth register of all births in Iceland. **Births included:** live births and stillbirths $\geq$ 22 weeks, or birthweight $\geq$ 500 grams. **Number of live births per year:** 4,500 [29]Landspitali University Hospital (LUH) in Reykjavik provides hospital-level maternity care for all of Iceland [32] and accounts for approximately 75% all births in Iceland [33]. **Neonatal care:** <br>• Babies >22 weeks: Neonatal intensive care unit at LUH <br>• Babies >34 weeks: Special care baby unit in Akureyri |
| | Mortality | Icelandic Death Register: Child deaths | 2003–2017 | Directorate of Health records of death. Includes main underlying cause of death. Contains child identifier. |
| | Secondary care | Hospital Drugs Register | 2010–2017 | Record of medications administered at LUH. Includes drug type, dosage, and date of administration. |
| | | Icelandic Patient Registers | 2003–2017 | • Maternal diagnoses from 12 months pre-pregnancy to 36 months post-partum <br>• Childhood neurodevelopmental and behavioural diagnoses from specialised care centres in Iceland where formal diagnoses made and care provided (State Diagnostic and Counselling Centre, Centre for Child Development and Behaviour, and LUH); includes primary care centres for neurodevelopmental disorders |
| ISRAEL | Pregnancy & birth | Rabin Medical Center Database | 2014–2019 | Hospital database of all births at Rabin Medical Center. **Births included:** live births and stillbirths $\geq$ 22 weeks. **Number of live births per year:** 9,500 at Rabin Medical Center (5% of 184,000 births per year in Israel in 2021 [34]) **Neonatal care:** <br>• Babies $\geq$ 23 weeks: Neonatal intensive care unit ("level 4") at Rabin Medical Center |
| NOVA SCOTIA | Pregnancy & birth | Nova Scotia Atlee Perinatal Database | 1990–2018 | Population birth register of all births in Nova Scotia. **Births included:** live births and stillbirths $\geq$ 20 weeks, or birthweight $\geq$ 500 grams. **Number of live births per year:** 7,400 [35] **Neonatal care:** <br>• Babies $\geq$ 23 weeks: Neonatal intensive care unit ("severe acuity, level 3") in Halifax <br>• Babies $\geq$ 32 weeks: "moderate acuity" ("level 2b") units in large regional centres [36] |

(Continued)

Table 1. (Continued)

| Region | Category | Data sources | Years available | Description of data |
|---|---|---|---|---|
| SCOTLAND | Pregnancy & birth | SMR02[a]: Scottish Maternity | 1997–2018 | Episode-level data which are collected any time a mother attends a hospital for an obstetric event. **Births included**: live births ≥ 22 weeks and stillbirths ≥ 24 weeks. **Number of live births per year**: 45,900 [31] **Neonatal care**: • Babies <27 weeks: intensive care units ("level 3") at tertiary hospitals • High dependency and special care (and lower volume of intensive care): local neonatal units ("level 2") • Special care units ("level 1") care for local populations [37] |
| | | SMR11[a]: Neonatal | 1996–2003 | Records of neonatal data only for unwell babies or babies with congenital anomalies. Contains child identifier. |
| | | SBR: Scottish Birth Record | 2000–2019 | Web-based system of electronic records of babies' neonatal care in Scotland, including antenatal information, post-delivery, readmissions and transfer. All births in Scotland from 2003 are registered on SBR within 2–3 days of birth. SBR was piloted in 2000 and superseded SMR11 completely in 2003. Contains child identifier. |
| | | National Records of Scotland: Stillbirths | 1997–2018 | Records of statutory registration of a stillbirth. Contains mother identifier. |
| | | Scottish Stillbirth and Infant Death Survey | 1997–2012 | Data on stillbirths, neonatal deaths (early and late), and, from 2013 onwards, late fetal losses (22 to 23 weeks). Dataset is now managed by MBRRACE-UK, commissioned by Healthcare Quality Improvement Partnership. Contains mother identifier. |
| | | Aberdeen Maternity and Neonatal Databank | 1997–2018 | Hospital database of all pregnancy events occurring at the Aberdeen Maternity Hospital (the only maternity hospital in Aberdeen) [38]. Includes 99% of births in Aberdeen [38]. **Number of live births per year**: 5,000 (11% of all live births in Scotland) [31] |
| | Mortality | National Records of Scotland: Child deaths & Adult deaths | 1997–2019 | Records of statutory registration of death, including the cause of death. Records provide information on whether deaths occurred during pregnancy/childbirth, within 42 days of termination of pregnancy (maternal deaths), or more than 42 days but less than 1 year after termination of pregnancy (late maternal deaths) [39, 40]. Child death records contain child identifier; adult death records contain mother identifier. |
| | Primary Care | Child Health Systems Programme Pre-School system | 1991–2019 | Electronic system which supports the delivery of Child Health Programme through automated invitations for preschool children to undergo screening, data collection from assessments of health and wellbeing, and provision of health promotion advice. Contains child identifier. |
| | Secondary care | SMR01[a]: General Acute | 1997–2019 | Episode-level data for patients discharged from inpatient or outpatient attendances at general or acute specialties. Contains child identifier. |
| | | SMR04[a]: Mental health | 1998–2019 | Episode-level data for patients from inpatient or outpatient attendances at psychiatric hospitals or units, or facilities caring for people with learning difficulties. Contains child identifier. |
| | | SMR06[a]: Cancer Registry | 1997–2019 | Data on all new diagnoses of cancer in Scottish residents. Includes information on treatment modalities used. Contains child identifier. |

For each region, numbers of live births per year are from 2020, unless otherwise specified, and are rounded to nearest hundred. "Episode-level data" refers to data recorded every time a patient interacts with the health care system (an "episode" of care), irrespective of duration.

LUH = Landspitali University Hospital. MBRRACE-UK = Mothers and Babies: Reducing Risk through Audits and Confidential Enquires across the UK.

[a] SMR = Scottish Morbidity Records, from Public Health Scotland.

**Table 2. Variables in the Co-OPT ACS cohort and data availability from contributing regions.**

| Category | Variable | Data availability from regions |
|---|---|---|
| **Pregnancy characteristics** | Maternal age at delivery | All |
| | Parity | All |
| | Total previous stillbirths | Finland, Nova Scotia, Scotland only |
| | Total previous neonatal deaths | Finland, Nova Scotia, Scotland only |
| | Plurality | All |
| | Maternal Body Mass Index at booking | All [a] |
| | Maternal height at booking | All [a] |
| | Maternal weight at booking | All [a] |
| | Maternal diabetes mellitus (pre-existing) | All |
| | Maternal hypertension (pre-existing) | All |
| | Marital status | Finland, Iceland, Nova Scotia, Scotland only |
| | Maternal ethnicity | Nova Scotia, Scotland only |
| | Maternal socioeconomic status | Scotland only |
| | Maternal smoking status at booking | Finland, Nova Scotia, Scotland only |
| | Assisted conception | Finland, Iceland, Israel, Nova Scotia only |
| | Pre-eclampsia | All |
| | Gestational diabetes mellitus | All |
| | Antepartum Haemorrhage | Finland, Iceland, Nova Scotia, Scotland only |
| | Chorioamnionitis (clinical and/or histological diagnosis) | All |
| | Antenatal hospital attendances | Finland, Nova Scotia only |
| | Antenatal hospital admissions | Nova Scotia, Scotland only |
| **Antenatal corticosteroids** | Any ACS administration | All |
| | Timing of ACS administration | Nova Scotia, Scotland (Aberdeen) only [b] |
| **Labour and birth outcomes** | Onset of labour | All |
| | Prelabour rupture of membranes | All |
| | Gestational age at birth | All |
| | Year of birth | All |
| | Month of birth | All |
| | Mode of birth | All |
| | Birthweight | All |
| | Birth length | Finland, Iceland, Nova Scotia, Scotland only |
| | Baby head circumference | Finland, Iceland, Nova Scotia, Scotland only |
| | Baby sex | All |
| | Congenital anomaly | All |
| | Apgar score at 5 minutes | All |
| | Stillbirth | All |
| | Early neonatal death (0–6 days) | Finland, Iceland, Nova Scotia, Scotland only |
| | Late neonatal death (7–27 days) | Finland, Nova Scotia, Scotland only |
| | Neonatal unit admission | All |
| | Level of Neonatal care provided | Scotland only |
| | Duration of Neonatal unit admission | Finland, Scotland only |

(*Continued*)

**Table 2.** (Continued)

| Category | Variable | Data availability from regions |
|---|---|---|
| **Maternal outcomes** | Maternal mortality | Finland, Nova Scotia, Scotland only |
| | Maternal admission to intensive/high dependency care | Finland only |
| **Child health & developmental outcomes** | Infant death | Finland, Iceland, Nova Scotia, Scotland only |
| | Newborn hearing screening failure | Scotland only |
| | Concern raised at first visit (undertaken at approximately 10 days old) | Scotland only |
| | Concern raised at various preschool developmental assessments undertaken at 6–8 weeks to 4–5 years old | Scotland only |
| | Preschool orthoptic visual screening failure | Scotland only |
| | Diagnoses of childhood neurodevelopmental disorders [c] | Finland, Iceland only |
| | Diagnoses of childhood physical and mental disorders (from hospital inpatient admissions and outpatient attendances) | Finland only |

See also S1 Table.

All = data for specific variable were available from all regions in the cohort (Finland, Iceland, Israel, Nova Scotia and Scotland). at booking = the first antenatal appointment. ACS = Antenatal corticosteroids.

[a] For births in Iceland, maternal Body Mass Index, height, and weight at booking were only available from 2012 onwards.

[b] Date of ACS administration were provided by Iceland, but since only year and month of birth were provided, it was not possible to determine gestational age at ACS administration or treatment-to-delivery interval.

[c] Diagnoses of childhood neurodevelopmental disorders were available up to 12 years old for births in Finland and up to 8 years old for births in Iceland.

for approximately half of all SMR02 discharges [31]. Until 2019, SMR02 did not include data on homebirths (1% of births in Scotland [27]) or births at non-NHS hospitals, unless data were subsequently recorded by an NHS hospital. However, SMR02 delivery records are consistently reported to be 99% complete for all known births registered by the statutory National Records of Scotland (NRS) per year [31]. The type of neonatal care provided in Scotland is location-dependent.

The Rabin Medical Center Database contains data from all births within this tertiary hospital alone, which accounts for 5% of births in Israel, and provides neonatal intensive care for babies from 23 weeks onwards.

**ACS exposure.** Details of any ACS exposure were provided in a basic nominal format (Yes / No / Unknown) for births in Finland, Israel, and Scotland. Birth data from Iceland were linked with hospital data on ACS administration, which included date of exposure, formulation, and number of doses of ACS used. Of note, for births in Iceland, ACS exposure data were only available from 2010 onwards, so births before 2010 were excluded from the Co-OPT ACS cohort. Additionally, as only month and year of birth was provided for births in Iceland, it was not possible to determine treatment-to-delivery time intervals for ACS administration, nor gestational age at ACS exposure. Nova Scotia provided ACS data categorised as treatment-to-delivery time intervals for ACS administration (first dose), and if known, specified whether dexamethasone or betamethasone was administered. For a subset of 7.1% of ACS-exposed births in Scotland, additional detail on gestational age at ACS administration (first dose, in completed weeks) was supplied by the Aberdeen Maternity and Neonatal Databank, using unique patient identifiers to enable linkage with SMR02.

**Perinatal and neonatal data.** Perinatal and neonatal data are provided by all four birth registers and by the Rabin Medical Center hospital database. In addition, for Iceland and

Scotland, national death records were linked with birth data to determine key perinatal outcomes such as stillbirths and neonatal deaths and to provide information on the cause of death. In Scotland, additional neonatal datasets (SMR11 and SBR) were linked with SMR02 to provide detailed information on neonatal conditions diagnosed and on durations of different levels of neonatal care provided.

Further information on the linkage of birth and death registers and the methods used to assign birth outcomes to babies based on information from multiple datasets are described in S2 Text.

**Childhood data.** Data on infant deaths (within the first year of life) are provided within birth registers for Finland and Nova Scotia. Data are available for deaths up to 12 years of age in both Scotland and Iceland, by linkage of birth registers with national death registers. No data on infant or child deaths were available from the Rabin Medical Center database.

In addition to comprehensive pregnancy, maternity and neonatal data provided by each country, additional data on child health and development, and inpatient and outpatient hospital attendances, are available for Finland, Iceland, and Scotland, from various sources. Live births in Finland were linked to hospital records of inpatient admissions and outpatient attendances from 2006 to 2018, which provide information on a range of diagnoses, including neurodevelopmental disorders and physical conditions. In Iceland, live births were linked with childhood diagnoses of neurodevelopmental and behavioural disorders from 2010 to 2018, provided by patient registers from specialised centres which provide care, counselling, and follow-up for children with these conditions.

Scottish live births were linked with child health and development data up to 5 years old, supplied by the records from the Child Health Systems Programme (CHSP) Pre-School information system. The CHSP Pre-School system supports the delivery of the Child Health Programme, a universal health promotion programme offered to all children and their families, by facilitating the recording of health data obtained at a series of preschool reviews. These assessments are undertaken at prespecified milestones to evaluate children's health, growth, and development, and are typically conducted by health visitors (nurses or midwives with specialist knowledge and training in community public health nursing, who provide support and advice for all families until a child starts school). These record linkages will enable analysis of long-term neurodevelopmental and behavioural outcomes for a significant subset of children in the Co-OPT ACS cohort. In the future, Scottish Morbidity Records for episodes of care (day-case and inpatient) within acute specialties (SMR01), cancer services (SMR06) and mental health services (SMR04) will be linked with live births in Scotland in the Co-OPT ACS cohort, to enable evaluation of longer-term benefits and safety of ACS in children using ICD-defined (International Statistical Classification of Diseases and Related Health Problems) outcome criteria, such as childhood infection.

## Data cleaning, harmonisation and quality assurance

Baseline maternity data from all five regions underwent standard data cleaning processes which are described in detail in S1 Text. This included identification of birth episodes (SMR02 only), screening for data outliers and nonsensical values, application of inclusion criteria and data cleaning rules (as defined by the Co-OPT collaborators), de-duplication of records, assessment of data completeness, and quality assurance checks. These processes enabled our successful "harmonisation" of the five datasets, which was essential to allow direct comparison of data between countries and to unify the data into one robust cohort. S1 Table shows pregnancy and birth-related variables which were available across all datasets after data cleaning and harmonisation.

Datasets from Finland, Iceland, Nova Scotia and Scotland employed ICD-10 codes (International Statistical Classification of Diseases and Related Health Problems, Tenth Revision [41]), to record diagnoses of new and pre-existing maternal conditions and congenital anomalies. Additionally, Iceland provided information on mode of birth as ICD-10 codes. Definitions based on ICD-10 codes were created for maternal conditions, congenital anomalies, and modes of birth; a consensus agreement was made by the Co-OPT collaborators on definitions, which were then applied across the datasets. Detailed ICD-10 code definitions are provided in Tables A-F in S1 File.

### Data linkage

Linkage processes are described in detail in S2 Text. The linkage between all datasets contributing to the Co-OPT ACS cohort is illustrated in Fig 1, and S1 Fig outlines filtering, transformation, and linkage of the Scottish datasets.

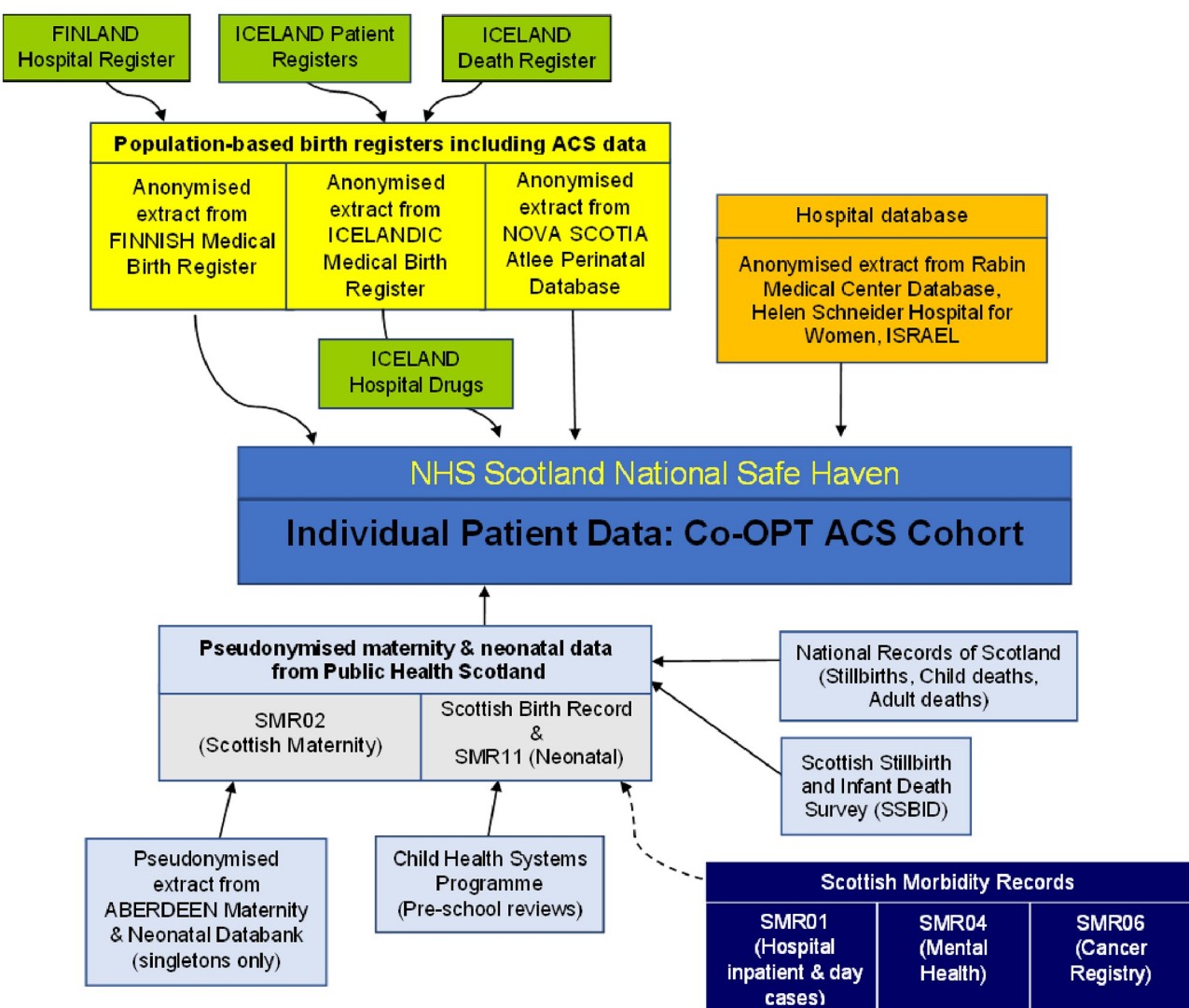

**Fig 1. Data flow diagram for Co-OPT ACS cohort.** ACS = Antenatal corticosteroids. NHS = National Health Service. SMR02 = Maternity Inpatient and Day Case Scottish Morbidity Records. SMR11 = Neonatal Scottish Morbidity Records.

### Ethics statement

This research has been conducted in accordance with the principles of the International Conference on Harmonisation Tripartite Guideline for Good Clinical Practice (ICH GCP). We obtained ethics approval from the ACCORD (Academic and Clinical Centre Office for Research and Development) Medical Research Ethics Committee (AMREC), which gave a favourable ethical opinion to the Co-OPT ACS study ("Benefits and Harms of Antenatal Corticosteroid Therapy (ACT)") on 19th December 2019, REC reference: 19-HV-063. The study was approved by the NHS Scotland Public Benefit and Privacy Panel (PBPP) for Health and Social Care (Reference 1718–0054) on 16th October 2019. PBPP approval includes access to data from Public Health Scotland as well as regulatory approvals to access the Nova Scotia Atlee Database, Icelandic Medical Birth Register, Finland Medical Birth Register, Israel Helen Schneider Hospital for Women Birth Cohort, Rabin Medical Center (Tel Aviv University) and Aberdeen Maternity & Neonatal databank (University of Aberdeen). Individual patient consent was not required as the study used anonymised and pseudonymised data.

## Results

### Co-OPT ACS cohort: Data completeness and characteristics

Rates of data completeness for key maternal and neonatal variables in the Co-OPT ACS cohort are shown in Table 3. Overall data completeness ranged from 85–100%, except for maternal Body Mass Index (BMI) at time of first antenatal appointment (67.2% complete), and detail surrounding timing of ACS administration (available for 15.5% ACS-exposed births).

There are 2,276,156 babies in the Co-OPT ACS cohort, from January 1990 to September 2019. Of all births, 2,210,368 (97.1%) were singleton pregnancies, and 92.9% were at term

**Table 3. Data completeness for key maternal and neonatal variables within Co-OPT ACS cohort, per region, and in the whole cohort (%).**

| KEY VARIABLE | Finland | Iceland | Israel | Nova Scotia | Scotland | All births in Co-OPT ACS cohort |
|---|---|---|---|---|---|---|
| Maternal age at delivery | 100.0 | 100.0 | 99.6 | 100.0 | 100.0 | 100.0 |
| Parity | 100.0 | 100.0 | 99.9 | 100.0 | 99.3 | 99.7 |
| Maternal BMI at booking [a] | 97.7 | 62.4 | 57.0 | 42.4 | 54.3 | 67.2 |
| Maternal smoking at booking | 97.1 | N/A | N/A | 67.3 | 92.5 | 87.8 |
| Maternal pre-existing diabetes | 100.0 | 100.0 | 100.0 | 100.0 | 71.9 | 85.4 |
| Maternal pre-existing hypertension | 100.0 | 100.0 | 100.0 | 100.0 | 100.0 | 100.0 |
| ACS exposure | 100.0 | 100.0 | 100.0 | 100.0 | 78.8 | 89.0 |
| Timing of ACS exposure [b,c] | N/A | N/A | N/A | 98.1 | 7.1 | 15.5 |
| Gestational age at birth | 100.0 | 100.0 | 100.0 | 100.0 | 100.0 | 100.0 |
| Mode of birth | 100.0 | 100.0 | 96.5 | 100.0 | 100.0 | 99.9 |
| Apgar score at 5 minutes | 87.4 | 100.0 | 99.9 | 99.1 | 98.2 | 94.8 |
| Birthweight | 100.0 | 100.0 | 100.0 | 99.9 | 99.9 | 99.9 |

N/A = Not available in dataset. BMI = Body Mass Index. "at booking" = the first antenatal appointment. ACS = Antenatal corticosteroids.

[a] For Nova Scotia, maternal BMI pre-pregnancy was provided.

[b] Timing of ACS administration for births in Scotland has been supplied by linkage of SMR02 data with data from Aberdeen Maternity and Neonatal Databank, on the date the first dose of ACS was given (from which gestation at administration has been derived). For births in Nova Scotia, timing of ACS administration has been provided in a categorical format (categories include given < 24 hours before delivery, given 24 to ≤ 48 hours, > 48 hours but < 7 days before delivery, >7 days before delivery, unknown when given). For births in Iceland, as only month and year of birth was provided, it was not possible to determine exact timing of ACS administration in relation to gestational age at exposure, or interval between exposure and birth.

[c] Data completeness for timing of ACS exposure has been calculated as a percentage of ACS-exposed babies, for each region and the whole cohort.

(from 37 completed weeks onwards). The characteristics of the Co-OPT ACS cohort are presented in Table 4.

## Quality assurance

For SMR02, published quality assurance literature and data completeness information provided by PHS confirmed that overall rates of data completeness were appropriate; this was used to inform temporal patterns of missing data [42]. In addition, we compared numbers of live births and stillbirths per year in Scotland within the Co-OPT ACS cohort (defined by rules outlined in S2 Text) with published annual live birth and stillbirth statistics from PHS and from NRS. These aligned closely, which confirmed high data quality for Scottish births in the cohort and validated the rules we created to define stillbirths using both SMR02 and NRS.

## Descriptive data: International and temporal trends

**Gestational age at birth.** The median gestational age at birth in the Co-OPT ACS cohort was 40 (interquartile range 2.0) weeks. The overall PTB rate (as defined by the World Health Organization: number of liveborn preterm births before 37 completed weeks of gestation divided by total number of live births [43]) in the Co-OPT ACS cohort was 6.9%. Rates of PTB across the countries ranged from 5.6% in Finland to 8.6% in Israel. Fig 2A and 2B show categorised gestational ages at birth (live births and stillbirths) across regions in the Co-OPT ACS cohort, for singleton and multiple births, respectively.

**Mode of birth.** Data on mode of birth were harmonised to enable classification into five categories: vaginal, assisted vaginal, planned Caesarean birth, unscheduled Caesarean birth, and other/unspecified. Fig 3 shows trends in mode of birth across time, and across regions in the Co-OPT ACS cohort.

**ACS exposure.** Rates of ACS administration were determined for each region that contributed to Co-OPT, and were evaluated over time, and by gestational age at birth, as shown in Fig 4A–4D. Data on ACS exposure were complete for births from all countries except Scotland, where 21.2% of births had missing ACS data. When births with missing ACS data were excluded, 3.6% (72,490) of births in the cohort had been exposed to ACS. Rates of ACS exposure were highest in Scotland (4.3% of births) and lowest in Iceland and Israel (2.1% of births). Of all babies in the cohort who received ACS, 26.8% were born at term. Sixty-seven percent of singleton births before 34 weeks received ACS (19,239 births), compared to 77.9% of multiple births (7945 births).

Further information on timing of ACS administration was only available for 15.5% of all ACS-exposed births in the cohort (98.1% ACS-exposed births in Nova Scotia and 7.1% ACS-exposed births in Scotland). ACS data from Nova Scotia were categorised based on time intervals between ACS exposure (first dose) and birth, as shown in Fig 5. Almost two thirds of ACS-exposed babies in Nova Scotia (5476 babies, 64.9%) were born over 7 days after ACS had been administered.

Through linkage with data from the Aberdeen Maternity and Neonatal Databank, detail on gestational age at administration of the first dose of ACS was available for births in Aberdeen, as shown in Fig 6. Of the ACS-exposed babies, over one third of babies (n = 963, 34.1%) received ACS at late preterm gestations (from 34 until 37 completed weeks), and 12.6% of babies received ACS from 37 weeks onwards.

**Longitudinal childhood data.** For all births in Finland, follow-up data were available in the form of 5.9 million linked outpatient hospital records for 604,876 (81.2%) babies and 525,099 linked inpatient hospital records for 263,947 (35.4%) babies. Hospital records included a range of primary and secondary diagnoses, as well as dates of attendances, and duration of

**Table 4. Characteristics of the Co-OPT ACS cohort (n = 2,276,156).**

| | Finland | Iceland | Israel | Nova Scotia | Scotland | Co-OPT ACS cohort (All babies) |
|---|---|---|---|---|---|---|
| **Total number of babies (% of Co-OPT ACS cohort)** | 742,919 (32.6) | 34,542 (1.5) | 43,826 (1.9) | 272,060 (12.0) | 1,182,809 (52.0) | 2,276,156 |
| **Total number of mothers** | 434,459 | 25,439 | Unknown [a] | 157,653 | 698,985 | Unknown |
| **Years of birth** | 2006–2018 | 2010–2017 | 2014–2019 | 1990–2018 | 1997–2018 | 1990–2019 |
| **Characteristic** | Number of babies, from region (%), unless specified | | | | | Total number of babies (%), unless specified |
| **Maternal age[b] (years)** | | | | | | |
| <20 | 15,049 (2.0) | 785 (2.3) | 137 (0.3) | 17,319 (6.4) | 77,732 (6.6) | 111,022 (4.9) |
| 20–24 | 109,062 (14.7) | 5625 (16.3) | 3741 (8.6) | 54,783 (20.1) | 206,887 (17.5) | 380,098 (16.7) |
| 25–29 | 229,115 (30.8) | 11,093 (32.1) | 11,205 (25.7) | 85,341 (31.4) | 320,161 (27.1) | 656,915 (28.9) |
| 30–34 | 242,179 (32.6) | 10,234 (29.6) | 15,436 (35.3) | 77,856 (28.6) | 352,587 (29.8) | 698,292 (30.7) |
| 35–39 | 119,260 (16.1) | 5514 (16.0) | 9985 (22.9) | 31,668 (11.6) | 187,411 (15.8) | 353,838 (15.5) |
| ≥ 40 | 28,254 (3.8) | 1291 (3.7) | 3168 (7.3) | 5088 (1.9) | 38,021 (3.2) | 75,813 (3.3) |
| Missing | 0 | 0 | 154 | 5 | 19 | 178 |
| **Plurality** | | | | | | |
| Singleton | 721,868 (97.2) | 34,010 (98.5) | 42,661 (97.3) | 264,403 (97.2) | 1,147,426 (97.0) | 2,210,368 (97.1) |
| Twin | 20,686 (2.8) | 524 (1.5) | 1133 (2.6) | 7443 (2.7) | 34,638 (2.9) | 64,424 (2.8) |
| Higher-order multiple | 365 (0.0) | 8 (0.0) | 32 (0.1) | 214 (0.1) | 745 (0.1) | 1364 (0.1) |
| Missing | 0 | 0 | 0 | 0 | 0 | 0 |
| **Parity** | | | | | | |
| Primiparous | 309,145 (41.6) | 13,740 (39.8) | 14,117 (32.2) | 122,353 (45.0) | 527,452 (44.9) | 986,807 (43.5) |
| Multiparous | 433,653 (58.4) | 20,802 (60.2) | 29,687 (67.8) | 149,683 (55.0) | 647,574 (55.1) | 1,281,399 (56.5) |
| Missing | 121 | 0 | 22 | 24 | 7783 | 7950 |
| **Maternal BMI (kg/m$^2$)[c]** | | | | | | |
| Underweight (<18.5) | 26,139 (3.6) | 603 (2.8) | 2077 (8.3) | 5435 (4.7) | 17,766 (2.8) | 52,020 (3.4) |
| Normal (18.5–24.9) | 446,333 (61.5) | 11,296 (52.4) | 15,367 (61.5) | 56,556 (49.1) | 308,944 (48.1) | 838,496 (54.8) |
| Overweight (25–29.9) | 161,122 (22.2) | 5267 (24.4) | 4991 (20.0) | 27,543 (23.9) | 180,326 (28.1) | 379,249 (24.8) |
| Obese (30–39.9) | 83,081 (11.4) | 3497 (16.2) | 2380 (9.5) | 21,151 (18.4) | 117,953 (18.4) | 228,062 (14.9) |
| Severely obese (≥40) | 9476 (1.3) | 891 (4.1) | 176 (0.7) | 4569 (4.0) | 16,884 (2.6) | 31,996 (2.1) |
| Missing | 16,768 | 12,988 | 18,835 | 156,806 | 540,936 | 746,333 |
| **Maternal smoking[d]** | | | | | | |
| No | 613,745 (85.1) | 0 | 0 | 144,915 (79.1) | 849,834 (77.6) | 1,608,494 (80.5) |
| Yes | 107,590 (14.9) | 0 | 0 | 38,313 (20.9) | 244,813 (22.4) | 390,716 (19.5) |
| Missing | 21,584 | 34,542 | 43,826 | 88,832 | 88,162 | 276,946 |
| **ACS exposure** | | | | | | |
| No | 720,451 (97.0) | 33,806 (97.9) | 42,908 (97.9) | 263,462 (96.8) | 892,258 (95.7) | 1,952,885 (96.4) |
| Yes | 22,468 (3.0) | 736 (2.1) | 918 (2.1) | 8598 (3.2) | 39,770 (4.3) | 72,490 (3.6) |
| Missing | 0 | 0 | 0 | 0 | 250,781 | 250,781 |
| **Gestational age at birth (completed weeks)** | | | | | | |

*(Continued)*

**Table 4.** (Continued)

|  | Finland | Iceland | Israel | Nova Scotia | Scotland | Co-OPT ACS cohort (All babies) |
|---|---|---|---|---|---|---|
| < 28 | 2559 (0.3) | 117 (0.3) | 284 (0.6) | 1261 (0.5) | 5276 (0.4) | 9497 (0.4) |
| 28–31 | 4020 (0.5) | 171 (0.5) | 350 (0.8) | 2006 (0.7) | 10,571 (0.9) | 17,118 (0.8) |
| 32–36 | 36,499 (4.9) | 1769 (5.1) | 3213 (7.3) | 17,601 (6.5) | 76,038 (6.4) | 135,120 (5.9) |
| 37–38 | 131,258 (17.7) | 5568 (16.1) | 13,769 (31.4) | 65,335 (24.0) | 218,430 (18.5) | 434,30 (19.1) |
| 39–41 | 536,083 (72.2) | 26,208 (75.9) | 25,989 (59.3) | 177,057 (65.1) | 841,156 (71.1) | 1,606,493 (70.6) |
| 42–45 | 32,500 (4.4) | 709 (2.1) | 221 (0.5) | 8800 (3.2) | 31,338 (2.6) | 73,568 (3.2) |
| Missing | 0 | 0 | 0 | 0 | 0 | 0 |
| **Mode of birth** | | | | | | |
| Vaginal | 551,989 (74.3) | 26,529 (76.8) | 31,042 (73.4) | 180,006 (66.2) | 723,189 (61.2) | 1,512,755 (66.5) |
| Assisted vaginal | 65,241 (8.8) | 3727 (10.8) | 3535 (8.4) | 26,354 (9.7) | 144,647 (12.2) | 243,504 (10.7) |
| Planned Caesarean | 49,500 (6.7) | 2107 (6.1) | 5883 (13.9) | 30,776 (11.3) | 131,855 (11.2) | 220,121 (9.7) |
| Unscheduled Caesarean | 76,180 (10.3) | 2177 (6.3) | 1843 (4.4) | 34,879 (12.8) | 182,697 (15.5) | 297,776 (13.1) |
| Missing | 9 | 2 | 1523 | 45 | 421 | 2000 |
| **Baby sex** | | | | | | |
| Male | 379,994 (51.2) | 17,704 (51.3) | 21,379 (48.8) | 139,354 (51.2) | 606,402 (51.3) | 1,164,833 (51.2) |
| Female | 362,889 (48.8) | 16,386 (48.7) | 22,446 (51.2) | 132,688 (48.8) | 576,154 (48.7) | 1,111,013 (48.4) |
| Missing or undetermined | 36 | 2 | 1 | 18 | 253 | 310 |
| **Stillbirth** | | | | | | |
| No | 740,700 (99.7) | 34,456 (99.8) | 43,688 (99.7) | 270,866 (99.6) | 1,177,358 (99.5) | 2,267,068 (99.6) |
| Yes | 2219 (0.3) | 86 (0.2) | 138 (0.3) | 1194 (0.4) | 5451 (0.5) | 9088 (0.4) |
| Missing | 0 | 0 | 0 | 0 | 0 | 0 |
| **Apgar score at 5 minutes** | | | | | | |
| Median (IQR) | 9.0 (1.0) | 10.0 (1.0) | 10.0 (0.0) | 9.0 (1.0) | 9.0 (0.0) | 9.0 (1.0) |
| Missing | 93,838 | 0 | 23 | 2427 | 21,752 | 118,040 |
| **Birthweight (g)** | | | | | | |
| Median (IQR) | 3525 (660.0) | 3640 (694.0) | 3216 (621.0) | 3455 (700.0) | 3410 (720.0) | 3455 (700.0) |
| Missing | 290 | 13 | 14 | 260 | 1647 | 2224 |

Data are presented as total numbers of babies (% of available data), except Apgar scores at 5 minutes and birthweights, which are presented as median values (interquartile range). BMI = Body Mass Index. ACS = Antenatal corticosteroids. IQR = Interquartile range.

[a] No maternal identifiers available for Israel.

[b] Maternal age at delivery.

[c] Maternal BMI at time of booking (first antenatal appointment).

[d] Maternal smoking status at the time of booking (first antenatal appointment).

inpatient admissions. Most babies could be linked with multiple hospital episodes (some children had up to 19 outpatient attendances). The oldest age at follow-up for Finnish children was 12 years.

In total, 859 (2.5%) Icelandic live births from the cohort were linked with diagnoses of mental, behavioural, and neurodevelopmental disorders, made between 2010 to 2017, after referral to specialist centres. For children born in Iceland, the oldest age at diagnosis was 8 years old. A range of diagnoses were recorded in the form of 37 ICD-10 codes (F01-F99), and included

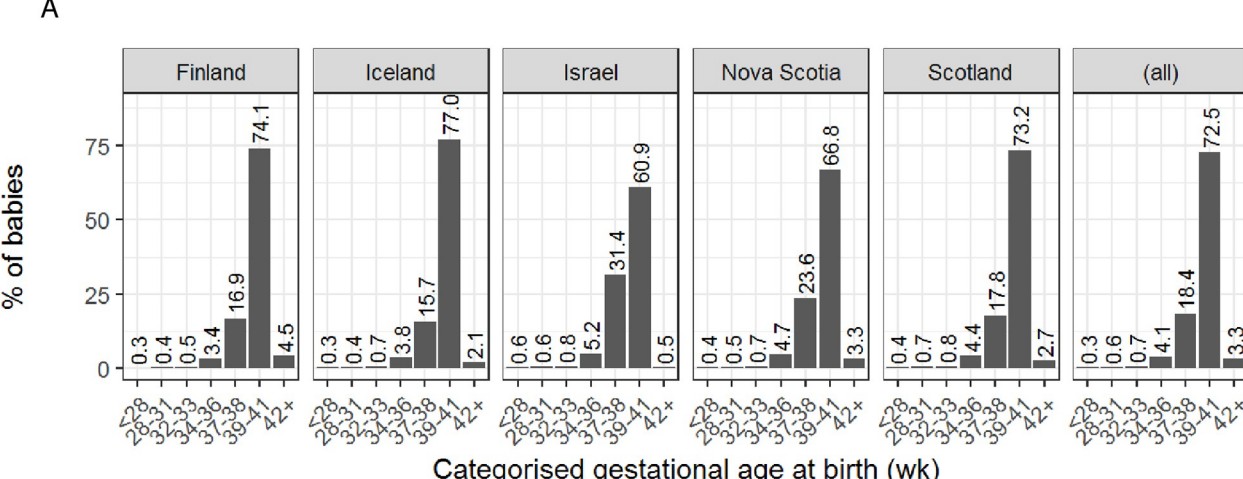

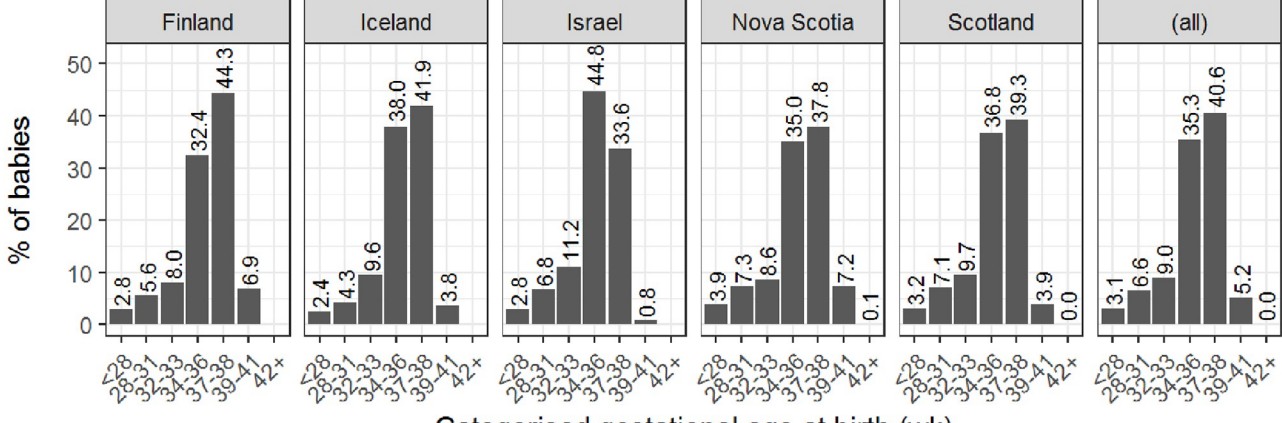

**Fig 2. Gestational age at birth in the Co-OPT ACS cohort.** (A) Gestational age at birth in singletons births. (all) = All singleton babies in the Co-OPT ACS cohort (live births and stillbirths). (wk) = completed weeks of gestation. (B) Gestational age at birth in multiple births. (all) = All multiple babies in the Co-OPT ACS cohort (live births and stillbirths). (wk) = completed weeks of gestation.

affective disorders, intellectual disabilities and behavioural or emotional disorders, such as attention deficit hyperactivity disorder.

Of the 99% of Scottish live births in the SMR02 which were provided with a Child Index (identifier necessary to enable linkage with further Scottish datasets), up to 88.9% (1.04 million babies) were linked with at least one review from the CHSP Pre-School system, up to 5 years old. This represents 45.8% of all live births in Co-OPT. Table 5 shows rates of CHSP Pre-School review coverage for live born children from SMR02 with Child Indexes. As some reviews at specific milestones were discontinued and replaced during the cohort period, we grouped developmental reviews between 6–8 weeks to 2 years of age, and those over 2 years up to 4–5 years of age, with the exclusion of the pre-orthoptic visual screening assessment, which has been considered separately.

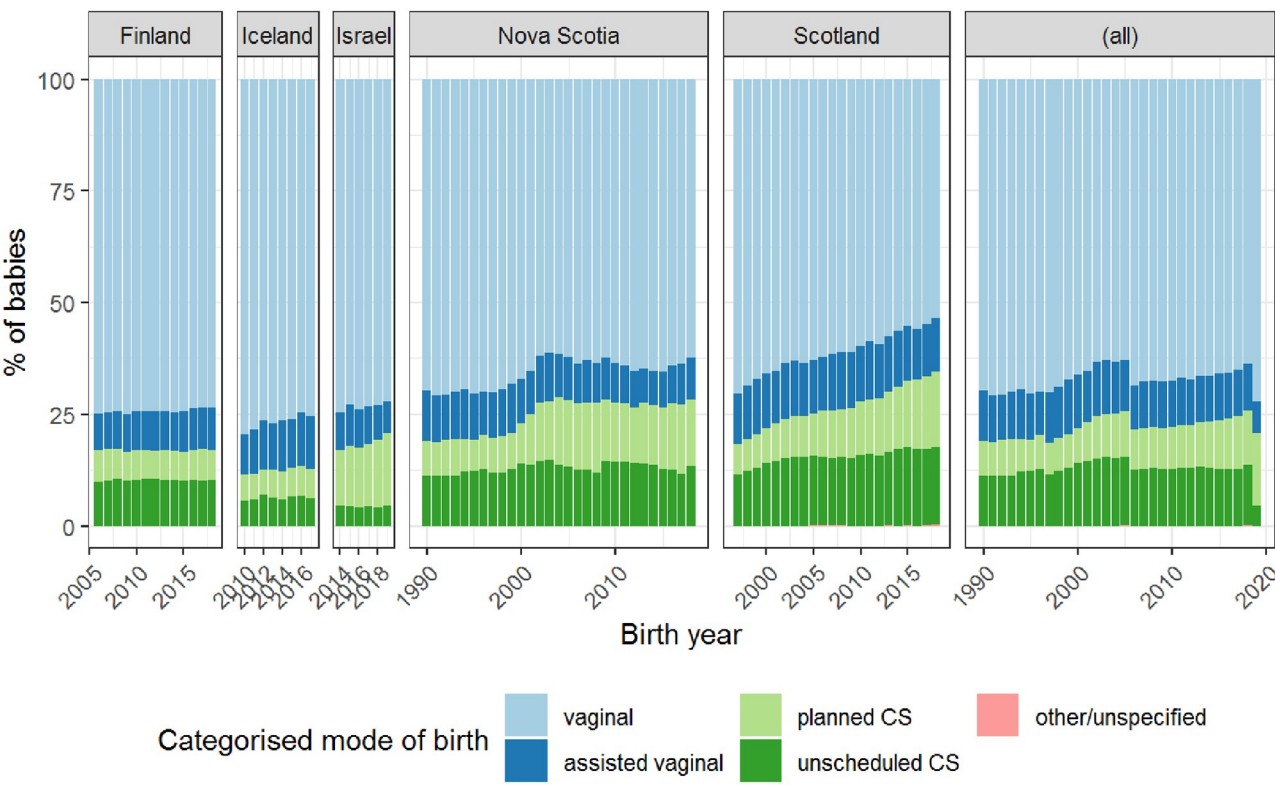

**Fig 3. Modes of birth in the Co-OPT ACS cohort over time.** Includes live births and stillbirths, and singleton and multiple births. The acute rise in proportion of vaginal births in the whole cohort ("all") in 2006 reflects births in Finland joining the Co-OPT ACS cohort. The final year in the whole cohort panel (2019) contains data from Israel alone. CS = Caesarean section. (all) = All babies in the Co-OPT ACS cohort.

## Discussion

### Key findings

In this large, international birth cohort, we have demonstrated that rates of ACS exposure were reasonably similar across all five regions, ranging from 2.1% to 4.3% of all births, and, as expected, the rates had gradually increased over the period studied. In births before 34 weeks, rates of ACS exposure in Iceland (43.3%) and Israel (31.2%) were lower than in Scotland (79.3%), Nova Scotia (63.3%) and Finland (62.2%). Given similar ACS prescribing practice across countries in pregnancies below 34 weeks (see S2 Table), this variation in preterm ACS exposure likely reflects differences in sources providing ACS data. For example, in Iceland, ACS exposure was provided by the drug register from the sole hospital providing hospital-level maternity care in Iceland (Landspitali University Hospital (LUH), see Table 1), where women are transferred in the context of imminent PTB, but this will potentially miss cases where the first dose of ACS has been administered elsewhere in Iceland and PTB occurs either before arrival at LUH or before administration of the second dose, and data from Israel reflects practice in a single hospital alone. This contrasts with the three other regions, where ACS data reflect practice across several hospitals across each country/province. This variation in the nature and context of ACS data provided across regions will need to be taken into consideration in future studies when interpreting outcomes associated with ACS exposure between countries. Importantly, most babies born before 34 weeks had received ACS (67.0% of singletons and 77.9% of multiples). While it is reassuring that rates of ACS exposure were reasonably

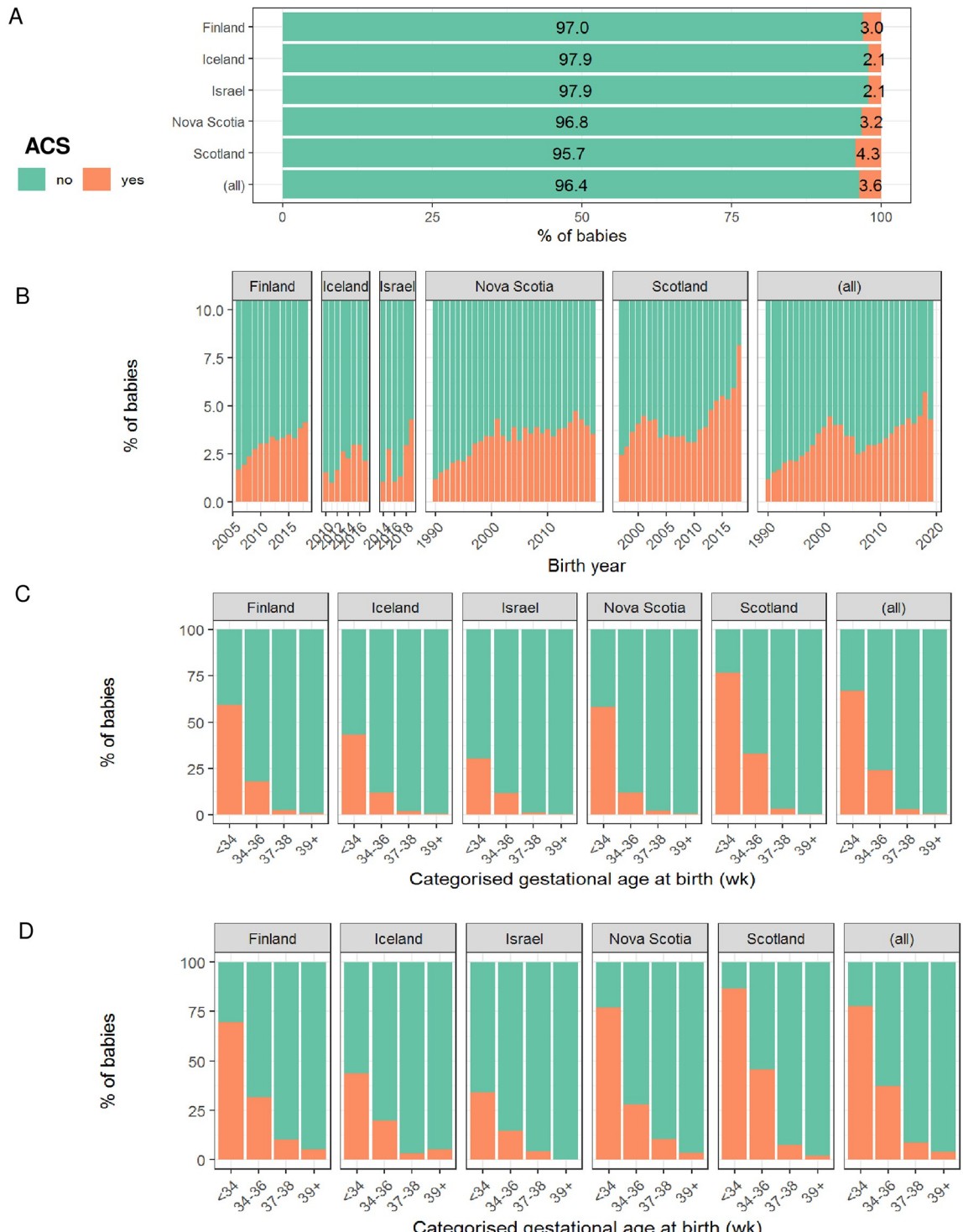

**Fig 4. Rates of ACS exposure in the ACS cohort.** (A) Overall rates of ACS exposure in the Co-OPT ACS cohort. (B) Rates of ACS exposure in the Co-OPT ACS cohort over time. (C) Rates of ACS exposure in singleton births in the Co-OPT ACS cohort, based on gestational age at birth. (D) Rates of ACS exposure in multiple births in the Co-OPT ACS cohort based on gestational age at birth. Fig 4A-4D include live births and stillbirths. Singleton and multiple births are presented in Fig 4A-4C includes singleton births only and Fig 4D includes multiple births only. Cases with missing ACS data were excluded. (wk) = completed weeks of gestation. (all) = All babies in the Co-OPT ACS cohort. ACS = exposed to antenatal corticosteroids.

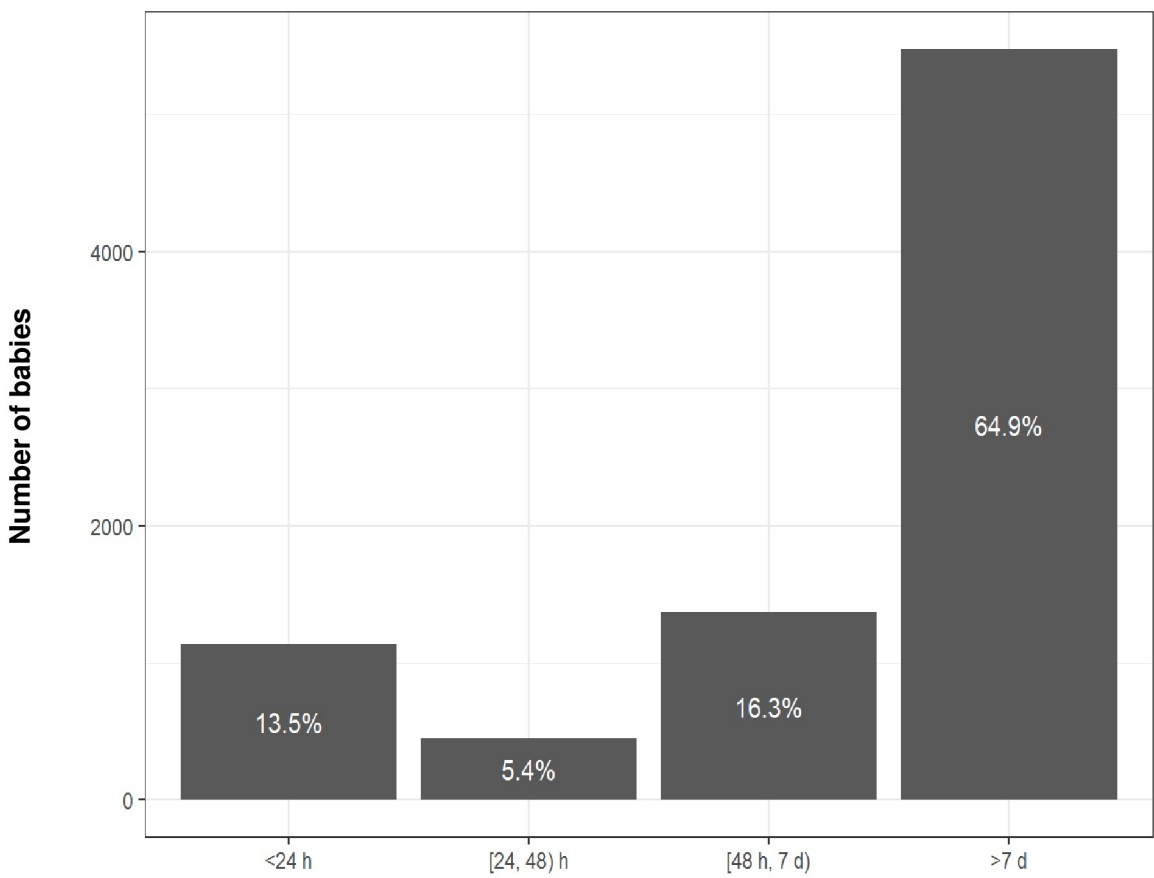

**Fig 5. ACS-exposed births in Nova Scotia categorised by time intervals between ACS administration and birth.** h = hours. d = days.

high amongst preterm births, undoubtedly there is scope to improve the identification of pregnancies which would benefit most from ACS administration.

The Nova Scotia birth data on treatment-to-delivery intervals showed that almost two thirds of ACS-exposed babies (64.9%, n = 5476) were born over 7 days after ACS administration, by which stage the therapeutic benefits have likely dissipated [16]. This particularly high rate of babies undelivered more than 7 days after ACS exposure exceeds rates found in previous studies [21, 44], and is a cause for concern. The introduction of policy changes and implementation strategies, such as the British Association of Perinatal Medicine's "Antenatal Optimisation Toolkit", will increase awareness of the need to optimise timing of ACS administration [45].

The practice of ACS administration at late preterm and term gestations remains controversial in view of its association with neonatal hypoglycaemia [46], and the paucity of data on its effects on childhood and adult outcomes [47]; indeed, guidelines on the administration of ACS beyond 34 weeks gestation vary across countries contributing to the Co-OPT ACS cohort, and over time, as shown in S2 Table. However, our findings indicate this has become common clinical practice in Scotland: almost half of ACS-exposed babies born in Aberdeen (46.8%, n = 1320) received ACS from 34 weeks onwards. In addition, of all term-born babies (from 37 weeks onwards) in the Co-OPT ACS cohort with ACS data available, 19,410 babies (1.0%) had been

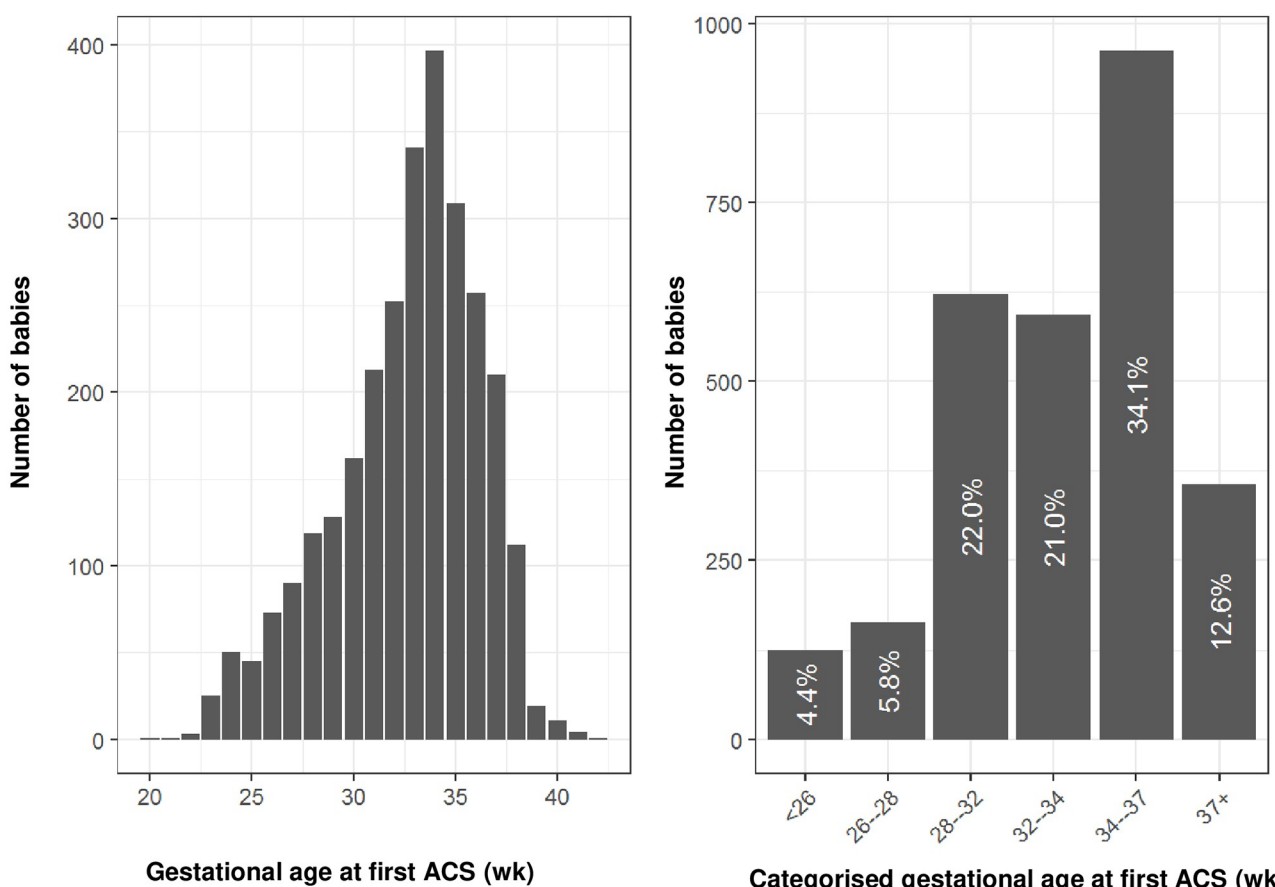

**Fig 6. Gestational age at ACS administration (first dose) for subset of Scotland births in Aberdeen, and categorised by gestational age groups.** (wk) = completed weeks of gestation.

exposed to ACS at some stage of the pregnancy, and 26.8% of all ACS-exposed babies in the cohort were born at term. This is disconcerting, particularly as large cohort studies have found associations in term-born infants between ACS exposure and increased risk of mental and behavioural disorders [9], and increased childhood utilisation of healthcare for suspected

**Table 5. Rates of Child Health Systems Programme Pre-School reviews for live births from SMR02 (with Child Indexes).**

| CHSP Pre-School Review | Dates of available data | Number of live borns from SMR02 reviewed | % all live borns from SMR02 (with CIndex) reviewed |
|---|---|---|---|
| Neonatal Hearing screen | 2001–2019 | 673,984 | 57.7 |
| First visit (10 days of age) | 2005–2019 | 1,037,657 | 88.9 |
| At least one developmental review after First visit, up to age 2[a] | 1997–2020 | 1,025,968 | 87.9 |
| At least one developmental review after age 2, up to age 5[b] | 1997–2020 | 520,742 | 44.6 |
| Pre-orthoptic Visual Screening (aged 4–5 years, preschool) | 2004–2019 | 507,317 | 43.5 |

CIndex = Child Index. CHSP = Child Health Systems Programme.

[a] Includes reviews at 6–8 weeks, 8–9 months, 13–15 months, 21–24 months, and two years of age.

[b] Includes reviews at 27–30 months, 39–42 months, and 4–5 years of age.

neurocognitive and neurosensory disorders [10]. Although data on timing of ACS were only available for a minority of births in the cohort, the practice of routinely administering ACS before planned, early-term (37 until 39 completed weeks) Caesarean birth applied only to Scotland, from 2010 onwards, after this was recommended in guidance by the Royal College of Obstetricians and Gynaecologists in 2010 ("Green-top guideline No. 7, Antenatal corticosteroids to reduce neonatal morbidity and mortality", published in 2010 and archived in 2016 [15]). As this has not been routine practice in any of the other regions studied (see S2 Table), we must therefore assume that the majority of term-born, ACS-exposed babies in the Co-OPT ACS cohort had received ACS earlier in pregnancy and did not deliver preterm: these babies had been given unnecessary treatment. This reiterates the need to improve the targeted use of ACS. The Co-OPT Consortium aims to develop predictive models to optimise timing of ACS prescription.

The rate of PTB within the Co-OPT ACS cohort was 6.9%, which is consistent with published rates from developed countries [48]. The rates of early-term births were noticeably higher in Israel and Nova Scotia than in other regions. With the exception of Finland, rates of vaginal birth across countries had gradually decreased over time, and there was a steady increase in planned Caesarean births in Israel, Nova Scotia and Scotland.

### Longitudinal childhood data

Linkage of 1.64 million live births with longitudinal childhood data from Scotland, Iceland and Finland, and associated details of ACS exposure, will enable robust evaluation of long-term childhood outcomes associated with ACS exposure, with a key focus on neurodevelopment. It is recognised that coverage of Scottish child health reviews gradually declines as the age of children advances [49], which is reflected by data in the Co-OPT ACS cohort in Table 5. Additionally, it is essential to recognise that rates of review coverage calculated in Table 5 do not account for neonatal or infant deaths that occurred before scheduled reviews, nor for children moving away from Scotland or outside regions that offered specific reviews at a certain time (information about which was not available in the cohort). Therefore, while the coverage of CHSP Pre-School reviews in the Co-OPT ACS cohort decreased with age, the overall review rates in Table 5 may underestimate the rates of follow-up, as these have not been restricted to eligible children alone.

Icelandic follow-up data on diagnoses of childhood neurodevelopmental disorders were limited to children in the cohort who reached an age at which symptoms and signs of these disorders had developed within the time studied (the oldest possible age of diagnosis being 8 years), yet several key neurodevelopmental disorders are typically diagnosed later in childhood. Additionally, diagnoses were limited to children who had not moved away from Iceland since birth.

Follow-up data in the cohort for babies born in Finland were more extensive than Iceland in terms of scale (81.2% and 35.4% babies linked with outpatient and inpatient records, respectively), and breadth of both physical and mental diagnoses, but are similarly limited by duration of follow-up available, which decreased the later the babies entered the cohort. The oldest possible age of diagnosis for children in Finland was 12 years. Furthermore, mental and neurodevelopmental disorders recorded in the Finnish Hospital Register represent the more severe spectrum of cases which require treatment at specialised facilities, and milder forms of these conditions will not be included.

### Strengths and limitations of the Co-OPT ACS cohort

The Co-OPT ACS cohort has several unique strengths. To our knowledge, the Co-OPT ACS cohort is the largest database of individual patient data on ACS use, containing 2.28 million

pregnancies and births from five regions. The cohort has patient representation from three continents, and its large scale enables robust evaluation of rarer outcomes, such as perinatal mortality. The use of observational data allows closer generalisability to "real-life" practice, particularly as RCTs on ACS are frequently restricted to specific subgroups of patients, with a particular focus on babies born preterm. In total, 72.5% (1.64 million) live births in the Co-OPT ACS cohort have longitudinal data available in the form of community preschool child health reviews, or diagnoses of physical and mental conditions in outpatient and inpatient hospital settings.

The Co-OPT ACS cohort benefits from regular consultation with cross-disciplinary researchers in the Consortium to reach a consensus agreement on appropriate definitions of variables, on the exclusion of data because of incompleteness or unreliability, and to determine approaches to data analysis, thereby maximising the potential of the Co-OPT Collaboration.

As clinical guidelines on ACS administration between 24–34 weeks' gestation were similar across countries in the cohort (see S2 Table), and largely consistent with guidance from the World Health Organization [16], these cohort findings can be generalised to other high-income countries, which have comparable guidelines on use of ACS.

In contrast to the majority of trials included in the Cochrane systematic review on ACS which recruited specific antenatal populations (particularly pregnant women with ruptured membranes, at gestational ages above 26 weeks, without reporting discrete outcomes for women with multiple pregnancies [1]), the Co-OPT ACS cohort includes births from 22 weeks onwards and reflects true clinical practice, without restriction to pregnancies with specific characteristics. Additionally, the Co-OPT ACS cohort includes 65,788 babies from multiple pregnancies, which will enable stratification of outcomes from ACS based on plurality, and robust subgroup analyses of the effects of ACS in singleton versus multiple pregnancies.

The 2020 Cochrane review on ACS presented a post-hoc analysis of ACS outcomes based on gestational age at trial entry, but this was only stratified into two overlapping categories, $\leq$ 35 weeks' and $\geq$ 34 weeks' gestation [1]. In contrast, the scale of the Co-OPT ACS cohort enables stratification of perinatal outcomes associated with ACS exposure by different gestational ages at birth. This includes extremely preterm births (before 28 completed weeks), which is particularly important given the lowering thresholds of perinatal viability and related guidance that ACS administration can be considered as early as 22 weeks [9, 50], and term births, which is pertinent given increasing evidence suggesting potential associations of ACS exposure with adverse neurodevelopmental outcomes in term-born babies [9, 10].

Quality assurance undertaken by comparing published numbers of stillbirths in Scotland from statutory NRS data with numbers of stillbirths in the Co-OPT ACS cohort (classified from linked SMR02 and NRS based on rules in S2 Text) was reassuring, as it demonstrated high rates of concordance between the two sources, ranging from 72 to 96% coverage of all statutory reported stillbirths.

The main limitation of the Co-OPT ACS cohort is missing data, particularly for ACS exposure in Scotland, where 21.2% of births had missing ACS data. Information analysts at PHS confirmed that ACS data completeness in SMR02 ranged from 46.5% to 73.1% during the first decade after the data field was introduced in 1997, after which it significantly improved, but they were not aware of any reason for this trend. Of note, ACS exposure is not a mandatory data field in SMR02 and it includes the option to record ACS status as "unknown" [51]. Data were limited for some other key variables, such as maternal ethnicity (available only for Scottish data), BMI (data completeness only 67.2% overall), and smoking history, on which data were unavailable for Iceland or Israel. It is likely these will be important covariates in analyses of ACS exposure on key outcomes. Where appropriate, multiple imputation techniques will be applied to missing variables within datasets in future analyses based on the Co-OPT ACS

cohort. The impact of missing data on the robustness of conclusions drawn from the Co-OPT ACS cohort will be carefully evaluated in attrition analyses. Finally, the lack of maternal identifier data from Israel prevents the identification of babies born to the same mother and precludes our determination of the total number of pregnancies in the cohort; this information is available for all other regions, however, which will enable important causal inference techniques such as sibpair analyses.

Another limitation of the Co-OPT ACS cohort is the lack of information available on either gestational age at exposure to ACS, or on time intervals between ACS administration and birth. Only Nova Scotia provided data on treatment-to-delivery interval, which represents only 11.6% of all ACS-exposed babies in the cohort, but information on treatment-to-delivery intervals is required to make clinically meaningful comparisons of optimal and suboptimal timing of ACS exposure. Data on gestational age at ACS administration were provided for over 90,000 Scottish singleton births from the Aberdeen Maternity and Neonatal Databank, but this was provided as completed weeks of gestation, which prevented determination of precise intervals between ACS exposure and birth. Additionally, not all babies in the Aberdeen Databank had baby identifiers, which prevented their linkage with data in SMR02. In total, birth records with ACS timing data from the Aberdeen Maternity and Neonatal Databank were linked with 7.1% of ACS-exposed births in Scotland. Similarly, although Iceland provided dates of ACS administration, only month and year of birth were available, precluding determination of clinically meaningful intervals between ACS exposure and birth. Missing data on the timing of ACS exposure increases likelihood of bias from residual confounding, as the effects of ACS exposure are intricately linked with the timing of ACS administration. Finally, only Iceland provided the number of courses of ACS given in each pregnancy. The increased use of electronic hospital prescribing systems, such as the national implementation of "HEPMA" (Hospital Electronic Prescribing & Medicines Administration) across NHS Scotland [52], should overcome these limitations by generating rich sources of high quality data on ACS administration, including detail of doses, formulation and timing. There will be potential for such data to be linked with maternity, neonatal and childhood data, which can be used to improve knowledge of the safety and benefits of ACS.

The scale of longitudinal follow-up data available in the Co-OPT ACS cohort is a major strength of the cohort, as it will enable robust evaluation of associations between ACS exposure and childhood outcomes, which will form the focus of future reports from Co-OPT. However, the variation in follow-up data between regions is a study limitation, as this reduces the potential to harmonise outcomes across datasets in the cohort. For example, the evaluation of neurodevelopmental outcomes is limited to children of preschool age born in Scotland, and children born in Iceland and Finland who have been formally diagnosed with neurodevelopmental diagnoses in a hospital or specialist setting, up to the age of 8 and 12 years, respectively. This precludes the evaluation of effects of ACS exposure on developmental or behavioural concerns which are milder, more subtle, or which do not manifest until later in childhood / adolescence / adulthood, and on educational outcomes. These are major areas of interest in the context of potentially unnecessary ACS treatment [9, 53]. Furthermore, given the heterogeneity of neurodevelopmental outcome data from these countries, ongoing analyses of possible associations of ACS exposure with early child neurodevelopment are limited to single centre analyses of data from Scotland alone. The contribution of data from different regions in Co-OPT to analyses of long-term outcomes will vary based on the availability and comparability of longitudinal data available for each outcome. The Co-OPT ACS cohort contains data from birth registers, but health and gender inequalities cause major disparities in birth registration worldwide [54], so there is a degree of selection bias in the cohort as it only represents countries that are able to provide this high-quality data. All countries that contributed data to the cohort are within the top 20 countries globally with the highest Human Development Index [55], yet the majority of preterm babies are born in

low- and middle-income countries [56]. Therefore, caution should be exercised when making inferences about the risks and benefits of ACS in lower- and middle-income countries.

In conclusion, through linkage and harmonisation of population-level individual patient data, we have created the Co-OPT ACS cohort, which is the largest international birth cohort to date comprising data on ACS exposure and on maternal, perinatal and childhood outcomes. The Co-OPT ACS cohort contains 2.28 million pregnancies and births, from five regions in three continents, and spans almost thirty years. Its use of routinely collected observational data enables closer generalisability of findings to "real-life" clinical practice. We demonstrated that 3.6% of all babies in the cohort had been exposed to ACS at some stage during the pregnancy. Concerningly, 26.8% of ACS-exposed babies had been born from 37 weeks onwards, underlining the urgent need to improve prediction of PTB and to design policies to highlight the need to optimise timing of ACS administration. Optimal delivery of ACS to preterm infants significantly improves perinatal outcomes. There is, however, rising concern about possible associations of potentially unnecessary ACS exposure with increased risk of adverse perinatal, childhood and maternal outcomes [6, 15]. The large scale and international representation of the Co-OPT ACS cohort will enable robust assessment of ACS exposure on rare outcomes, such as perinatal mortality, and longitudinal follow-up will provide comprehensive evaluation of the short- and long-term safety of ACS.

## Supporting information

**S1 Text. Data cleaning processes.**
(PDF)

**S2 Text. Data linkage processes.**
(PDF)

**S1 Table. Key maternity and neonatal variables in Co-OPT ACS cohort available across all contributing datasets after data cleaning and harmonisation.** ACS = Antenatal corticosteroids. M = Main deciphering variable. C/E = Confounder/Effect Modifier. O = Outcome. $\sqrt{}^*$ = available using ICD-10 codes and definitions agreed by Co-OPT collaborators (described in Tables A-F in S1 File).
(PDF)

**S2 Table. Overview of key recommendations on use of antenatal corticosteroids across regions included in the Co-OPT ACS cohort.** ACS = Antenatal Corticosteroids. mg = milligrams. NICU = Neonatal Intensive Care Unit. NICE = National Institute for Health and Care Excellence. SOGC = Society of Obstetricians and Gynaecologists of Canada. SMFM = Society for Maternal-Fetal Medicine. MDT = Multidisciplinary Team. RCOG = Royal College of Obstetricians & Gynaecologists. [1] The latest version of the relevant guideline (**"Primary source of guidelines"**) published **during the years when births in the Co-OPT ACS cohort occurred in that region** (region-dependent), has been cited for each different source of guidelines, for each region, and changes in guidance over the time studied has been summarised in the table. Please note that some cited guidelines have been archived since publication and have been superseded by updated versions, which were published after the last birth in the Co-OPT ACS cohort from that region occurred. [2] if high risk of imminent medically indicated or spontaneous preterm birth within the subsequent 7 days. $\sqrt{}$ = routine use of antenatal corticosteroids recommended. ($\sqrt{}$) = consider use of antenatal corticosteroids. **X** = routine use of antenatal corticosteroids not recommended.
(PDF)

**S1 Fig. Filtering, transformation and linkage of Scottish datasets to create "linked Scottish dataset".** SMR02 = Maternity Inpatient and Day Case Scottish Morbidity Records. eDRIS = electronic Data Research and Innovation Service (Public Health Scotland). MotherID / MIndex = unique mother identifier. CIndex = unique child identifier (child ID). NRS = National Records of Scotland. SMR11 = Neonatal Scottish Morbidity Records. SBR = Scottish Birth Record.
(TIF)

**S1 File. ICD-10 code definitions.** ICD-10 = International Classification of Diseases and Related Health Problems, 10th Revision.
(PDF)

## Acknowledgments

We are grateful to the Co-OPT collaborators from Finland, Iceland, Israel, Nova Scotia, and Scotland, who have provided high-quality patient data, without which the Co-OPT ACS cohort would not have been possible.

We acknowledge Public Health Scotland for providing us with a secure data analytical platform in which to undertake this research and are particularly grateful to Anna Schneider who has been the data controller for this project.

**Co-OPT collaborators**: Karel Allegaert (Belgium), Jasper Been (Netherlands), David Burgner (Australia), Sohinee Bhattacharya (UK), Kate Duhig (UK), Kristjana Einarsdóttir (Iceland), John Fahey (Canada), Lani Florian (UK), Abigail Fraser (UK), Mika Gissler (Finland), Cynthia Gyamfi-Bannerman (USA), Bo Jacobsson (Sweden), Eyal Krispin (Israel), Stefan Kuhle (Canada), Marius Lahti-Pulkkinen (Finland), Jessica Miller (Australia), Ben Mol (Australia), Sarah Murray (UK), Jane Norman (UK), Lars Henning Pedersen (Denmark), Richard Riley (UK), Devender Roberts (UK), Ewoud Schuit (Netherlands), Aziz Sheikh (UK), Ting Shi (UK), Joshua Vogel (Australia), Rachael Wood (UK), John Wright (UK), Helga Zoega (Australia).

## Author Contributions

**Conceptualization:** Rebecca M. Reynolds, Karel Allegaert, Jasper V. Been, Abigail Fraser, Mika Gissler, Kristjana Einarsdóttir, Lani Florian, Bo Jacobsson, Joshua P. Vogel, Helga Zoega, Sohinee Bhattacharya, Eyal Krispin, Lars Henning Pedersen, Devender Roberts, Stefan Kuhle, Ben W. Mol, David Burgner, Ewoud Schuit, Aziz Sheikh, Rachael Wood, Cynthia Gyamfi-Bannerman, Jessica E. Miller, John Wright, Sarah R. Murray, Sarah J. Stock.

**Formal analysis:** Emily M. Frier, Chun Lin, Sarah J. Stock.

**Funding acquisition:** Sarah J. Stock.

**Methodology:** Emily M. Frier, Chun Lin, Rebecca M. Reynolds, Karel Allegaert, Jasper V. Been, Abigail Fraser, Mika Gissler, Kristjana Einarsdóttir, Lani Florian, Bo Jacobsson, Joshua P. Vogel, Helga Zoega, Sohinee Bhattacharya, Eyal Krispin, Lars Henning Pedersen, Devender Roberts, Stefan Kuhle, John Fahey, Ben W. Mol, David Burgner, Ewoud Schuit, Aziz Sheikh, Rachael Wood, Cynthia Gyamfi-Bannerman, Jessica E. Miller, Kate Duhig, Marius Lahti-Pulkkinen, John Wright, Sarah J. Stock.

**Project administration:** Emily M. Frier, Chun Lin, Rebecca M. Reynolds, Sarah J. Stock.

**Software:** Chun Lin.

**Supervision:** Rebecca M. Reynolds, Sarah J. Stock.

**Writing – original draft:** Emily M. Frier, Chun Lin.

**Writing – review & editing:** Emily M. Frier, Chun Lin, Rebecca M. Reynolds, Karel Allegaert, Jasper V. Been, Abigail Fraser, Mika Gissler, Kristjana Einarsdóttir, Lani Florian, Bo Jacobsson, Joshua P. Vogel, Helga Zoega, Eyal Krispin, Lars Henning Pedersen, Devender Roberts, Stefan Kuhle, John Fahey, Ben W. Mol, David Burgner, Ewoud Schuit, Aziz Sheikh, Rachael Wood, Cynthia Gyamfi-Bannerman, Jessica E. Miller, Kate Duhig, Marius Lahti-Pulkkinen, Eran Hadar, Sarah J. Stock.

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
