## [Decision Letter · Decision Letter 0]

21 Oct 2022

PONE-D-22-23203Consortium for the Study of Pregnancy Treatments (Co-OPT): An international birth cohort to study the effects of Antenatal CorticosteroidsPLOS ONE

Dear Dr. Frier,

Thank you for submitting your manuscript to PLOS ONE. After careful consideration, we feel that it has merit but does not fully meet PLOS ONE’s publication criteria as it currently stands. Therefore, we invite you to submit a revised version of the manuscript that addresses the points raised during the review process.

We look forward to receiving your revised manuscript.

Kind regards,

Quetzal A. Class, PhD

Academic Editor

PLOS ONE

Journal Requirements:

Additional Editor Comments:

Dear Authors,

Your work documents the establishment of an important dataset that will likely have future use. It is understood that this type of manuscript is important in the publication of future work with this dataset, therefore, we hope that this can be published. However, given one of the reviewer's major concerns that your data have limitations and that the focus of this work is on presenting the dataset rather than on the associations between risk and outcome, we are asking that you spend time responding and describing why these concerns can be overcome. This invitation to revise is not a guarantee of publication, but rather an opportunity for you to address important limitations and caveats to the dataset. I would also like more detail on the variations on ACS administration/dosing/etc as if there is a causal association, these dosing and administration practices will alter the associations.

Thank you and we look forward to your revision.

Reviewers' comments:

Reviewer's Responses to Questions

**Comments to the Author**

1. Is the manuscript technically sound, and do the data support the conclusions?

Reviewer #1: Yes

Reviewer #2: Partly

2. Has the statistical analysis been performed appropriately and rigorously? 

Reviewer #1: Yes

Reviewer #2: N/A

3. Have the authors made all data underlying the findings in their manuscript fully available?

Reviewer #1: Yes

Reviewer #2: No

4. Is the manuscript presented in an intelligible fashion and written in standard English?

Reviewer #1: Yes

Reviewer #2: Yes

5. Review Comments to the Author

Reviewer #1: This paper describes the first study within Co-OPT focusing on antenatal corticosteroids (ACS) in five countries between 1990-2019, including more than 2 million births. The overall strengths include the large cohort to study several important and rare outcomes including perinatal mortality and also neurodevelopmental disorders later in childhood. The limitations with the overall Co-OPT ACS cohort, include large missing numbers for certain exposures, outcomes and covariates, however these are well-described in the discussion section. Nevertheless, this is something the authors will have to take into consideration for each specific subsequent study question using this data.

I believe that this manuscript may be of interest for the readers of PONE and useful in the future for subsequent papers using data from the Co-OPT, but I have some suggestions to improve the manuscript:

1. While the introduction gives a very comprehensive overview of the history of ACS and the knowledge gap, it becomes quite long and some parts may be shortened to keep the interest of the reader.

2. In the methods on page 11, the authors state that the clinical guidelines on use of ACS in the included countries are similar and with guidance from the WHO. The clinical guidelines could be summarized and described with similarities and differences in the appendix, and further acknowledged in the discussion.

3. In the methods on page 12, there is a referral to a URL for a dictionary. However, this URL is not included in the paper and now says “tbc”.

4. Throughout the paper the word “episode” is used. For what I understand, an episode corresponds to a hospital visit or a duration of time including when the woman gave birth or similar. Are you taking the number of days into consideration for these episodes, or is this word used to describe any visits within the health care system? Episode-level data is also mentioned in table 1, which to me is a bit unclear what it means.

5. On page 15, the author states that assessments undertaken to evaluate children’s health, growth etc are done by “health visitors”. Please define what a “health visitor” is.

6. To me it seems unclear that you could have as many pregnancies as births when you also include multiple births. It would be clearer if you described the number of pregnant individuals leading to x number of births. I assume that the problem is that the number of pregnant women in Israel was unknown. This should be acknowledged in the discussion.

7. In the results section on page 28, I would strongly suggest to add that only 15.5% of the whole cohort had available information on timing of ACS exposure.

8. In the discussion section on page 30 the authors discuss that varying rates of ACS exposure in the different countries “likely reflect variations in ACS prescribing practice”. To me this contradicts the previous statement that clinical practice is similar in the different countries. Please clarify both the clinical guidelines (as mentioned in point 2) and in the discussion how potential differences between the countries in prescribing patterns may affect future studies using data on ACS from the full cohort.

Reviewer #2: The authors promised to deal with short and long term effects of ACS on maternal and infant development. They present interesting data on (country-specific) ACS application. However, I have some general objections against the current structure/content of the manuscript.

To a large extend the manuscript deals with the description how they have linked population-based data sources from five regions to create the Co-OPT ACS cohort as a basis for the intended research including some characteristics of the cohort.

This basic information of the population to be studied should be summarized in the Method section of the manuscript and details should be described as supplement material.

Instead of methodological issues and basic characteristics the Result and Discussion section of the manuscript should present details of the observed maternal/infant outcome, i.e. potential effects of ACS instead. Also, the discussed strengths and limitations of the study would make more sense along with the findings of the study.

In particular, the problem of merging infant data from completely different systems to study rather subtle neurodevelopmental effects can only be judged in light of these clinical data. The difficulty to interpret any observed symptoms as related to ACS or prematurity (or other pregnancy issues) underscores the necessity to report infant outcomes. The author’s theoretical consideration of this problem is correct, but does not satisfy the reader’s interest.

Even more the author’s important goal “to improve the targeted use of ACS… and to develop predictive models to optimise timing of ACS prescription” only makes sense with the presentation of details of infant outcome. This is even more true considering that only Nova-Scotia data (11.6% of the total cohort) include precise information on gestational time of ACS exposure and exposure to delivery interval.

Actually, I am not yet convinced of the “unification of the data into a robust cohort” which consist mainly (approx. 85%) of Scottish and Finish data. On top of that, in 20% of cases of the largest sub-cohort (Scotland) ACS exposure is unknown.

All in all, the authors have started an important and interesting project. The manuscript, however, should be enriched with the (long term) clinical findings of their exposed study cohort in comparison to the non-exposed.

6. PLOS authors have the option to publish the peer review history of their article (what does this mean?). If published, this will include your full peer review and any attached files.

Reviewer #1: No

Reviewer #2: No

---

## [Author Response · Author response to Decision Letter 0]

30 Nov 2022

[We have uploaded the point-by-point responses to reviewer and editor comments in the document uploaded "Response to Reviewers", which includes our responses in bold. The content of this document is replicated below].

Reviewer #1: This paper describes the first study within Co-OPT focusing on antenatal corticosteroids (ACS) in five countries between 1990-2019, including more than 2 million births. The overall strengths include the large cohort to study several important and rare outcomes including perinatal mortality and also neurodevelopmental disorders later in childhood. The limitations with the overall Co-OPT ACS cohort, include large missing numbers for certain exposures, outcomes and covariates, however these are well-described in the discussion section. Nevertheless, this is something the authors will have to take into consideration for each specific subsequent study question using this data.

I believe that this manuscript may be of interest for the readers of PONE and useful in the future for subsequent papers using data from the Co-OPT, but I have some suggestions to improve the manuscript:

1. While the introduction gives a very comprehensive overview of the history of ACS and the knowledge gap, it becomes quite long and some parts may be shortened to keep the interest of the reader.

We thank the reviewer for this recommendation and have shortened the Introduction. 

2. In the methods on page 11, the authors state that the clinical guidelines on use of ACS in the included countries are similar and with guidance from the WHO. The clinical guidelines could be summarized and described with similarities and differences in the appendix, and further acknowledged in the discussion.

Thank you for this very helpful suggestion. We have added S2 Table in the Technical appendix, which summarises the clinical guidelines on use of ACS over the time studied, and we have cited this table in the Methods and the Discussion, as well as acknowledging important similarities and differences between guidelines in the Discussion.

3. In the methods on page 12, there is a referral to a URL for a dictionary. However, this URL is not included in the paper and now says “tbc”.

Thank you for highlighting this. The location of the data dictionary had not been confirmed at the time of original submission as it was pending confirmation of data sharing permissions from the University of Edinburgh. The data dictionary is now freely available through DataShare (University of Edinburgh) and the URL has been included in the manuscript (https://datashare.ed.ac.uk/handle/10283/4768) and the Supporting information S1 File. 

4. Throughout the paper the word “episode” is used. For what I understand, an episode corresponds to a hospital visit or a duration of time including when the woman gave birth or similar. Are you taking the number of days into consideration for these episodes, or is this word used to describe any visits within the health care system? Episode-level data is also mentioned in table 1, which to me is a bit unclear what it means.

To clarify the term “episode”, we have added in parentheses “any interaction with the health care system, irrespective of duration”, on page 11. We have also added a definition in the legend of Table 1 to clarify what episode-level data is: “Episode-level data refers to data recorded every time a patient interacts with the health care system (an “episode” of care), irrespective of duration”. 

5. On page 15, the author states that assessments undertaken to evaluate children’s health, growth etc are done by “health visitors”. Please define what a “health visitor” is.

We have added information in parentheses on page 14 which defines what a health visitor is: “nurses or midwives with specialist knowledge and training in community public health nursing, who provide support and advice for all families until a child starts school”. 

6. To me it seems unclear that you could have as many pregnancies as births when you also include multiple births. It would be clearer if you described the number of pregnant individuals leading to x number of births. I assume that the problem is that the number of pregnant women in Israel was unknown. This should be acknowledged in the discussion.

Thank you for drawing this to our attention. We were not provided with a maternal identifier for birth data from Israel, which does indeed prevent our identification of the total number of pregnant individuals in the cohort. We have acknowledged this limitation in the Discussion, within Strengths and Limitations, on page 36:“Finally, the lack of maternal identifier data from Israel prevents the identification of babies born to the same mother and precludes our determination of the total number of pregnancies in the cohort; this information is available for all other regions, however, which will enable important causal inference techniques such as sibpair analyses.”

In the Results, we now state the total number of singleton pregnancies, as well as the total number of babies, to clarify the number of pregnancies vs the number of births, as these are indeed different: “Of all births, 2,210,368 (97.1%) were singleton pregnancies, and 92.9% were at term (from 37 completed weeks onwards)” (page 21). 

7. In the results section on page 28, I would strongly suggest to add that only 15.5% of the whole cohort had available information on timing of ACS exposure.

Thank you for this suggestion. We had included this information in the Results section on page 21, which reviews data completeness along with overall characteristics of the cohort (“Overall data completeness ranged from 85-100%, except for… detail surrounding timing of ACS administration (available for 15.5% ACS-exposed births”). However, we have now modified the sentence on page 28 which refers to timing of ACS exposure as we agree it is very important to highlight the low availability of timing data when presenting these results: “Further information on timing of ACS administration was only available for 15.5% of all ACS-exposed births in the cohort (98.1% ACS-exposed births in Nova Scotia and 7.1% ACS-exposed births in Scotland)”.

8. In the discussion section on page 30 the authors discuss that varying rates of ACS exposure in the different countries “likely reflect variations in ACS prescribing practice”. To me this contradicts the previous statement that clinical practice is similar in the different countries. Please clarify both the clinical guidelines (as mentioned in point 2) and in the discussion how potential differences between the countries in prescribing patterns may affect future studies using data on ACS from the full cohort.

Thank you for highlighting these important points and for the valuable recommendations. We agree that the sentence quoted was misleading and contradicts the earlier statement that clinical practice is similar in different countries. We have therefore modified this section as follows, to explain more clearly how we interpret the variation in ACS exposure across regions, and citing the new table (S2 Table) included in the Supporting information, as per your suggestion in point 2. On page 31, we state: “Given similar ACS prescribing practice across countries in pregnancies below 34 weeks (see S2 Table), this variation in preterm ACS exposure likely reflects differences in sources providing ACS data. For example, in Iceland, ACS exposure was provided by the drug register from the sole hospital providing hospital-level maternity care in Iceland (LUH, see Table 1), where women are transferred in the context of imminent PTB, but this will potentially miss cases where the first dose of ACS has been administered elsewhere in Iceland and PTB occurs either before arrival at LUH or before administration of the second dose, and data from Israel reflects practice in a single hospital alone. This contrasts with the three other regions, where ACS data reflect practice across several hospitals across each country/province.”

We have also added the subsequent statement on page 31 which underlines the importance of taking this into consideration when interpreting results from future studies: “This variation in the nature and context of ACS data provided across regions will need to be taken into consideration in future studies when interpreting outcomes associated with ACS exposure between countries.”

In addition, in the discussion, we have acknowledged differences in ACS prescribing practices across regions by adding the following statement on page 31: “indeed, guidelines on the administration of ACS beyond 34 weeks gestation vary across countries contributing to the Co-OPT ACS cohort, and over time, as shown in S2 Table.” 

We have also modified the statement on administration of ACS at term gestations on page 32, and added a citation of the new S2 table, to improve interpretation of data on term-born babies exposed to ACS: “Although data on timing of ACS were only available for a minority of births in the cohort, the practice of routinely administering ACS before planned, early-term (37 until 39 completed weeks) Caesarean birth applied only to Scotland, from 2010 onwards, after this was recommended in guidance by the Royal College of Obstetricians and Gynaecologists in 2010 [48] (guidance archived in 2016 [15]). As this has not been routine practice in any of the other regions studied (see S2 Table), we must therefore assume that the majority of term-born, ACS-exposed babies in the Co-OPT ACS cohort had received ACS earlier in pregnancy and did not deliver preterm” 

Finally, we have amended the statement in the Strengths and limitations to specify that guidelines on ACS administration were similar for pregnancies between 24 to 34 weeks’ gestation (since the main variations between countries relate to ACS administration beyond 34 weeks), on page 34: “guidelines on ACS administration between 24-34 weeks’ gestation were similar across countries in the cohort (see S2 Table).

Reviewer #2: The authors promised to deal with short and long term effects of ACS on maternal and infant development. They present interesting data on (country-specific) ACS application. However, I have some general objections against the current structure/content of the manuscript.

To a large extend the manuscript deals with the description how they have linked population-based data sources from five regions to create the Co-OPT ACS cohort as a basis for the intended research including some characteristics of the cohort.

This basic information of the population to be studied should be summarized in the Method section of the manuscript and details should be described as supplement material.

Instead of methodological issues and basic characteristics the Result and Discussion section of the manuscript should present details of the observed maternal/infant outcome, i.e. potential effects of ACS instead. Also, the discussed strengths and limitations of the study would make more sense along with the findings of the study.

In particular, the problem of merging infant data from completely different systems to study rather subtle neurodevelopmental effects can only be judged in light of these clinical data. The difficulty to interpret any observed symptoms as related to ACS or prematurity (or other pregnancy issues) underscores the necessity to report infant outcomes. The author’s theoretical consideration of this problem is correct, but does not satisfy the reader’s interest.

Even more the author’s important goal “to improve the targeted use of ACS… and to develop predictive models to optimise timing of ACS prescription” only makes sense with the presentation of details of infant outcome. This is even more true considering that only Nova-Scotia data (11.6% of the total cohort) include precise information on gestational time of ACS exposure and exposure to delivery interval.

Actually, I am not yet convinced of the “unification of the data into a robust cohort” which consist mainly (approx. 85%) of Scottish and Finish data. On top of that, in 20% of cases of the largest sub-cohort (Scotland) ACS exposure is unknown.

All in all, the authors have started an important and interesting project. The manuscript, however, should be enriched with the (long term) clinical findings of their exposed study cohort in comparison to the non-exposed.

Thank you for your helpful comments and thorough review of the manuscript. However, with regard to the request that the manuscript should be enriched with the long-term clinical outcomes from the cohort, respectfully, we wish to indicate that providing outcome data was not intended to be within the scope of this initial paper, the purpose of which is to describe why the cohort was established, to outline how the cohort was created, and to present the characteristics, strengths, and limitations of the cohort. The Co-OPT Collaboration intends to use the Co-OPT ACS cohort to evaluate a large range of different outcomes, using robust outcome definitions which will be agreed by the Collaborators. As outcome data availability differs between countries included in the cohort, and the outcomes vary in nature, the evaluation of their associations with ACS exposure will require different analytical approaches and will be undertaken using different subsets of the cohort (outcome-dependent). This will form the basis of several future studies, which will have their own strengths and limitations, and we do not think it is feasible to condense all of this work into a single manuscript. The primary purpose of this paper is to outline the cohort. 

To address these issues further, we have included a statement in the Discussion on page 37, to acknowledge future work: “The scale of longitudinal follow-up data available in the Co-OPT ACS cohort is a major strength of the cohort, as it will enable robust evaluation of associations between ACS exposure and childhood outcomes, which will form the focus of future reports from Co-OPT.”

We have also modified the wording of the Abstract (“Its large scale will enable assessment of important rare outcomes”), the Methods (“These record linkages will enable analysis of…” ), and final two paragraphs of the introduction, to provide clarity on the purpose of the paper, and to acknowledge that the evaluation of outcomes associated with ACS will be undertaken in due course. Regarding the reference to our goal “to improve the targeted use of ACS… and to develop predictive models to optimise timing of ACS prescription”, we have added the word “ultimately” before the final sentence on predictive models, as this indeed will rely on detailed analysis of both short- and long-term outcomes, which is outside of the scope of this paper. 

Additional Editor Comments:

Dear Authors,

Your work documents the establishment of an important dataset that will likely have future use. It is understood that this type of manuscript is important in the publication of future work with this dataset, therefore, we hope that this can be published. However, given one of the reviewer's major concerns that your data have limitations and that the focus of this work is on presenting the dataset rather than on the associations between risk and outcome, we are asking that you spend time responding and describing why these concerns can be overcome. This invitation to revise is not a guarantee of publication, but rather an opportunity for you to address important limitations and caveats to the dataset. I would also like more detail on the variations on ACS administration/dosing/etc as if there is a causal association, these dosing and administration practices will alter the associations.

Thank you and we look forward to your revision.

Thank you for your comments and for the opportunity to revise the manuscript. 

We have responded to Reviewer #2’s concerns above, and as described in the point-by-point responses to comments from both Reviewers, we have attempted to highlight important limitations and caveats of the dataset in the discussion (including our responses to points 6 and 8 by Reviewer #1). 

Thank you for your important recommendation to include more detail on administration and dosing of ACS across the regions, as this will indeed alter associations of ACS exposure with outcomes. As per Reviewer #1’s recommendation, we have now included a table (S2 Table) which outlines variations in ACS administration practice across countries contributing to the Co-OPT ACS cohort, and we have cited this table in different parts of the Discussion to provide context to interpreting results (please see responses to point 2 and 8 by Reviewer #1).

Journal Requirements:

We have made alterations to meet all style requirements. All headings are now in sentence case, figure citations have been abbreviated to “Fig”, file names have been amended appropriately and Table 4 is now cell-based (embedded with Microsoft Excel). We have also removed the statement about the funding source from Acknowledgements and we have added the corresponding author’s initials in parentheses after her email address. 

We have also removed the Supporting information document (which included text, tables, figures and captions), and have instead uploaded these as separate files, with appropriate file names, to enhance navigation for the reader. All supporting information files are cited in the manuscript text, and captions are listed at the end of the manuscript file. 

We have addressed this by ensuring that the grant information provided is accurate, and that both grant numbers match: we note that the Funding Reference number for this study has been abbreviated from 209560/Z/17/A14/32 to 209560/Z/17 in some contexts, for example, in the published protocol (cited in the Methods), so to maintain consistency we have ensured that the abbreviated grant number is used throughout. 

As detailed in our Cover letter, the updated Financial Disclosure statement should read as follows: 

The Co-OPT ACS study is funded through a Wellcome Trust Clinical Career Development Fellowship grant (Funding Reference number 209560/Z/17) awarded to Sarah J Stock. The funders had no role in study design, data collection, data analysis, decision to publish or preparation of the manuscript. The Sponsor of the study is the University of Edinburgh (www.ed.ac.uk), Sponsor reference AC19119. For the purpose of open access, the author has applied a CC BY public copyright licence to any Author Accepted Manuscript version arising from this submission.

We have also wish to amend the “Competing interests” section, which should read as follows: Sarah J Stock is funded by a Wellcome Trust Clinical Career Development Fellowship (209560/Z/17) for research into antenatal corticosteroids. The funders had no role in study design, data collection and analysis, decision to publish or preparation of the manuscript. All other authors have declared that no competing interests exist.

Thank you for raising this important issue. As detailed in the Data Availability statement, we are unable to share our minimal data set publicly for this manuscript, which contains individual patient-level data, due to confidentiality and data protection requirements, and we have provided information for interested researchers to apply to gain access to the data on request. To provide further clarification, we have added the statement “as established by the individual data holders for data provided for each country, because the data set contains sensitive and potentially identifying data”. Of note, we have removed both the Data Availability statement and the Competing interests from the manuscript itself.

As detailed in our Cover letter, the updated Data Availability statement should now read as follows:

The Co-OPT ACS cohort is stored in the National Health Service (NHS) Scotland National Safe Haven, provided by Public Health Scotland (PHS) electronic Data Research and Innovation Service (eDRIS). Patient-level data underlying this article cannot be shared publicly due to data protection and confidentiality requirements, as established by the individual data holders for data provided by each country, because the data set contains sensitive and potentially identifying data. The Finnish National Institute for Health and Welfare, The Icelandic Directorate of Health, the Rabin Medical Center (Israel), The Reproductive Care Program of Nova Scotia, Public Health Scotland and National Records of Scotland are the data holders for the data used in this study. Data may be made available to approved researchers for analysis after securing relevant permissions from the data holders. Enquiries regarding data availability should be directed to Professor Mika Gissler, Finland (mika.gissler@thl.fi); Professor Kristjana Einarsdottir, Iceland (ke@hi.is); Dr Eyal Krispin, Israel (Eyalkrispin@gmail.com); Dr Stefan Kuhle, Canada (stefan.kuhle@dal.ca) and the eDRIS team at Public Health Scotland (phs.edris@phs.scot).

We have included the ethics statement in the Methods section of the manuscript, and in no other sections.

---

## [Decision Letter · Decision Letter 1]

17 Jan 2023

PONE-D-22-23203R1Consortium for the Study of Pregnancy Treatments (Co-OPT): An international birth cohort to study the effects of antenatal corticosteroidsPLOS ONE

Dear Dr. Frier,

Thank you for submitting your manuscript to PLOS ONE. After careful consideration, we feel that it has merit but does not fully meet PLOS ONE’s publication criteria as it currently stands. Therefore, we invite you to submit a revised version of the manuscript that addresses the points raised during the review process.

We appreciate your revisions and assert that the manuscript is close to being accepted.

We would like you to further qualify the manuscript by outlining the limitations of the dataset given the variation in outcomes. In particular, it is yet to be known that the combined dataset can disentangle time dependent effects of ANC from the multiple co-variates including prematurity related consequences due to missing of essential data and the heterogeneity of contributing institutions with their different focus on infant developmental endpoints. If the combined dataset does not have comparable endpoints of neurodevelopment, it may be more informative to perform single center studies to prevent problems of heterogeneity. Please go through the manuscript to adjust claims that are yet to be determined.

Thank you.

We look forward to receiving your revised manuscript.

Kind regards,

Quetzal A. Class, PhD

Academic Editor

PLOS ONE

Journal Requirements:

Reviewers' comments:

Reviewer's Responses to Questions

**Comments to the Author**

1. If the authors have adequately addressed your comments raised in a previous round of review and you feel that this manuscript is now acceptable for publication, you may indicate that here to bypass the “Comments to the Author” section, enter your conflict of interest statement in the “Confidential to Editor” section, and submit your "Accept" recommendation.

Reviewer #1: All comments have been addressed

Reviewer #2: (No Response)

2. Is the manuscript technically sound, and do the data support the conclusions?

Reviewer #1: Yes

Reviewer #2: No

3. Has the statistical analysis been performed appropriately and rigorously? 

Reviewer #1: Yes

Reviewer #2: N/A

4. Have the authors made all data underlying the findings in their manuscript fully available?

Reviewer #1: Yes

Reviewer #2: (No Response)

5. Is the manuscript presented in an intelligible fashion and written in standard English?

Reviewer #1: Yes

Reviewer #2: Yes

6. Review Comments to the Author

Reviewer #1: (No Response)

Reviewer #2: (No Response)

7. PLOS authors have the option to publish the peer review history of their article (what does this mean?). If published, this will include your full peer review and any attached files.

Reviewer #1: **Yes: **Anne Örtqvist

Reviewer #2: No

---

## [Author Response · Author response to Decision Letter 1]

8 Feb 2023

[We have uploaded a Cover letter, in which we have responded to the email we received from the journal staff. The content of this document is replicated below. Responses to comments from the Academic Editor are detailed in the uploaded document "Response to Reviewers"].

Dear Dr Class,

Thank you very much for the opportunity to further revise our manuscript “Consortium for the Study of Pregnancy Treatments (Co-OPT): An international birth cohort to study the effects of antenatal corticosteroids” (PONE-D-22-23203R1) for PLOS One.

We sincerely appreciate your helpful comments, which we have addressed when revising the manuscript. Our responses to these are provided in the document “Response to Reviewers”, in bold. We have also removed a reference which has since been archived, as detailed in that document. In keeping with with our initial revision of the manuscript, we have included below the updated versions of the Data Availability statement, Competing interests section and the Financial Disclosure statement. These statements also clarify the role of the funders.

We have uploaded a marked-up copy of the manuscript which highlights changes made to the original version, along with an unmarked version of the revised paper without tracked changes. All page numbers referred to in our responses refer to the marked-up copy of the manuscript.

We hope you find the revised manuscript has been strengthened as a result of these changes, and we look forward to your decision regarding publication.

Yours sincerely,

Emily Frier, on behalf of the co-authors

Updated Data Availability statement

The Co-OPT ACS cohort is stored in the National Health Service (NHS) Scotland National Safe Haven, provided by Public Health Scotland (PHS) electronic Data Research and Innovation Service (eDRIS). Patient-level data underlying this article cannot be shared publicly due to data protection and confidentiality requirements, as established by the individual data holders for data provided by each country, because the data set contains sensitive and potentially identifying data. The Finnish National Institute for Health and Welfare, The Icelandic Directorate of Health, the Rabin Medical Center (Israel), The Reproductive Care Program of Nova Scotia, Public Health Scotland and National Records of Scotland are the data holders for the data used in this study. Data may be made available to approved researchers for analysis after securing relevant permissions from the data holders. Enquiries regarding data availability should be directed to Professor Mika Gissler, Finland (mika.gissler@thl.fi); Professor Kristjana Einarsdottir, Iceland (ke@hi.is); Dr Eyal Krispin, Israel (Eyalkrispin@gmail.com); Dr Stefan Kuhle, Canada (stefan.kuhle@dal.ca) and the eDRIS team at Public Health Scotland (phs.edris@phs.scot).

Updated Competing interests section

Sarah J Stock is funded by a Wellcome Trust Clinical Career Development Fellowship (209560/Z/17) for research into antenatal corticosteroids. The funders had no role in study design, data collection and analysis, decision to publish, or preparation of the manuscript. All other authors have declared that no competing interests exist.

Updated Financial Disclosure statement

The Co-OPT ACS study is funded through a Wellcome Trust Clinical Career Development Fellowship grant (Funding Reference number 209560/Z/17) awarded to Sarah J Stock. The funders had no role in study design, data collection, data analysis, decision to publish, or preparation of the manuscript. The Sponsor of the study is the University of Edinburgh (www.ed.ac.uk), Sponsor reference AC19119. For the purpose of open access, the author has applied a CC BY public copyright licence to any Author Accepted Manuscript version arising from this submission.

---

## [Editor Report · Decision Letter 2]

16 Feb 2023

Consortium for the Study of Pregnancy Treatments (Co-OPT): An international birth cohort to study the effects of antenatal corticosteroids

PONE-D-22-23203R2

Dear Dr. Frier,

We’re pleased to inform you that your manuscript has been judged scientifically suitable for publication and will be formally accepted for publication once it meets all outstanding technical requirements.

Kind regards,

Quetzal A. Class, PhD

Academic Editor

PLOS ONE

Additional Editor Comments (optional):

Dear Authors,

Thank you for your continued dedication to this manuscript. We are accepting your manuscript.

Thank you,

Dr. Quetzal Class
---

## [Editor Report · Acceptance letter]

21 Feb 2023

PONE-D-22-23203R2 

Consortium for the Study of Pregnancy Treatments (Co-OPT): An international birth cohort to study the effects of antenatal corticosteroids 

Dear Dr. Frier:

I'm pleased to inform you that your manuscript has been deemed suitable for publication in PLOS ONE. Congratulations! Your manuscript is now with our production department. 

Kind regards, 

on behalf of

Dr. Quetzal A. Class 

Academic Editor

PLOS ONE